# Climatic impacts on mortality in pre-industrial Sweden

Tzu Tung Chen[1,*], Rodney Edvinsson[2], Karin Modig[3], Hans W. Linderholm[1], and Fredrik Charpentier Ljungqvist[4,5,6]

[1]Regional Climate Group, Department of Earth Sciences, University of Gothenburg, 413 90 Gothenburg, Sweden
[2]Department of Economic History and International Relations, Stockholm University, 106 91 Stockholm, Sweden
[3]Unit of Epidemiology, Institute of Environmental Medicine, Karolinska Institutet, 171 77 Stockholm, Sweden
[4]Department of History, Stockholm University, 106 91 Stockholm, Sweden
[5]Bolin Centre for Climate Research, Stockholm University, 106 91 Stockholm, Sweden
[6]Swedish Collegium for Advanced Study, Linneanum, Thunbergsvägen 2, 752 38 Uppsala, Sweden
[*]Current affiliation: The Public Health Agency of Sweden, 171 82 Solna, Sweden

**Correspondence:** Fredrik Charpentier Ljungqvist (fredrik.c.l@historia.su.se)

**Abstract.** Climate variability and change, as well as extreme weather events, have notable impacts on human health and mortality. In historical times, the effect of climate on health and mortality was stronger than today, owing to factors such as poor housing and healthcare along with that the nutrition status was meditated through climatic impacts on food production. Despite this, climatic impacts on mortality in the past remain poorly understood. This study aims to improve the understanding of climate effects on mortality, using annual mortality records and meteorological data from Sweden between 1749 and 1859. The analysis includes the entire population as well as subgroups based on sex and age. A statistically significant negative correlation was found between late winter and spring temperatures and mortality (i.e., lower temperatures = higher mortality and *vice versa*). We demonstrate that colder late winter and spring seasons were linked to higher mortality levels, not only for the same year but also the following year. Conversely, no statistically significant associations were observed between summer or autumn temperatures and mortality, and only weak associations existed with hydroclimate. The impact of late winter and spring season temperature on mortality was most pronounced for the same year in southern Sweden and during the 19th century, but stronger for the following year in central Sweden and during the 18th century. These findings call for further research, especially investigating specific diseases and additional contributing factors to the observed increase in mortality following cold late winter and spring seasons in Sweden during the late pre-industrial period.

## 1 Introduction

The effects of climate change on human health and mortality have gained increasing attention during recent years in response to emerging and projected threats from anthropogenic global warming (Semenza and Menne, 2009; van Daalen et al., 2022; Romanello et al., 2022). Climate change can have direct effects on health and mortality through changes in the frequency, duration and magnitude of exposure to temperature and hydroclimate extremes (Raymond et al., 2020; Calleja-Agius et al., 2021; Vicedo-Cabrera et al., 2021; Wu et al., 2022). Furthermore, climate change also influence the transmission of vector-borne diseases through an influence on pathogens, human susceptibility, and the abundance and distribution of certain hosts

and vectors (Mills et al., 2010; Carlton et al., 2016; Rocklöv and Dubrow, 2020; Liczbińska et al., 2024). In addition, but not least, the impact of adverse climate on poorer societies, especially during historical times, affects human nutritional status – and thus health and mortality – through its effects on agricultural productivity (Collet, 2019; Degroot et al., 2021; Ljungqvist et al., 2021). In particular, the effects of climate-triggered famine events on morbidity and mortality in the past have been comparatively well-studied during recent years (e.g., Slavin, 2016; Collet and Schuh, 2018; Huhtamaa et al., 2022; Ljungqvist et al., 2024). Various other direct and indirect climatic effects on human health and mortality in historical times have been reviewed elsewhere (e.g., Diaz et al., 2001; McMichael, 2012; Robbins Schug et al., 2023).

## 1.1 Climate–mortality relationships: past and present

Climate and weather extremes have had considerable effects on human health and mortality patterns, primarily through its influence on food production, in pre-industrial Europe (Pfister and Wanner, 2021), not the least in the Nordic countries (Huhtamaa and Ljungqvist, 2021). However, the climate–mortality relationship, and especially its geographical patterns, remains poorly quantified. Longer periods of colder temperatures tended to increase the general mortality (Galloway, 1986), colder winters in particular increased mortality among the elderly (Galloway, 1985). The effect of cold winters on increased mortality disappeared in England already by $c.$ 1800, whereas the increase in mortality in response to hot summers declined throughout Europe from the late 18th century onwards (Galloway, 1994). In a more recent study, Waldinger (2022) unveiled that in England from 1538 to 1838, warmer growing seasons were associated with lower subsequent mortality, and *vice versa*. This effect was larger in rural areas distant from major markets.

For Sweden, Eckstein et al. (1984) reported that higher January–March temperatures reduced mortality in 18th and 19th century, while the temperature effect was smaller or non-existent for the warmer months of the year. In Sweden, as was also the case in for example France, a statistically significant temperature effect on mortality prevailed all the way up to $c.$ 1900 (Galloway, 1994). For the city of Uppsala, east-central Sweden, Schumann et al. (2013) showed that over the 1749–1859 period higher spring temperature decreased mortality, while higher summer temperature instead increased mortality. Moreover, higher spring precipitation decreased mortality, while higher autumn precipitation increased mortality. Rocklöv et al. (2014a) conducted a similar study for Skellefteå parish, northern coastal Sweden, for the same period. They found increased mortality in response to colder winters and springs, and higher autumn precipitation, particularly among children aged 3–9, but not among infants. Åström et al. (2016) found that during the 19th and early 20th century higher temperatures as well as higher precipitation was associated with lower mortality in Skellefteå, but that the climate effect on mortality decreased over time. Another study has showed that higher temperatures during the summer months in Sweden over the 1880–1950 period was a significant factor in neonatal mortality rates, although the effect decreased over time attributable to improvements in healthcare and living conditions (Junkka et al., 2021). Perinatal mortality increased among the Swedish Sámi with cold winters (Schumann et al., 2019), and prior to $c.$ 1860 cold winter months increased neonatal mortality among the Sámi population (Karlsson et al., 2019). The magnitude of seasonal fluctuations in mortality rates among the elderly in Sweden has decreased substantially since at least the mid-19th century. The cohorts born in Sweden in 1800 had a risk of dying during the winter season that was almost

twice the risk of dying during the summer season, while the increased risk of dying during the winter season was only about 10 % for the cohorts born in Sweden in 1900 (Ledberg, 2020).

In contemporary Europe, the elderly tend to be most vulnerable to weather-related deaths. Cold spells are linked to higher excess mortality than heat waves (van Daalen et al., 2022). Current climate change projections, however, indicate that mortality attributed to heat will start to exceed the reduction of mortality attributed to cold during the second half of the 21st century (Martínez-Solanas et al., 2021). In line with this, a study from Switzerland found that over the past decades, population ageing has attenuated the decrease in cold-related mortality and amplified heat-related mortality (de Schrijver et al., 2022). Another study by Masselot et al. (2023), analysing non-optimal temperatures and mortality in urban populations across 854 European cities, found that vulnerability to temperature increased with age for both cold and heat – although the difference is less steep for heat than it is for cold – suggesting that the effect of heat affected all ages more homogeneously. Overall, the population aged older than 85 years accounted for 60 % of the total mortality burden of extreme temperatures.

Even if age is one of the most important risk factors of extreme temperature for mortality, other factors obviously modify the effect, e.g., housing standard and access to health care (Sera et al., 2019; Bakhtsiyarava et al., 2023). Rocklöv et al. (2014b) found that the effect on mortality by heat wave duration was modified by wealth, sex as well as health status. Summer temperature increases were associated with mortality increases in the group over 80 years as well as with mortality increases in groups with a previous myocardial infarction and with chronic obstructive pulmonary disease in the population younger than 65 years. During winter, excess mortality was found particularly in men and related to the duration of cold spells for the population older than 80.

Many, but not all, causes of death display seasonal patterns. The total number of deaths follows an annual cycle, with higher mortality during the winter months in the extra-tropical Northern Hemisphere including contemporary Sweden (Statistics Sweden, 2020). Pneumonia, influenza, chronic obstructive pulmonary disease (COPD), and other respiratory diseases, peak during the winter months (Lowen and Steel, 2014; Ballester et al., 2016; Achebak et al., 2023). A similar pattern, though less pronounced, is observed for cardiovascular diseases (Marti-Soler et al., 2014). Conversely, deaths from traffic accidents peak during summer months, while suicide and cancers show small or no seasonality (Rau et al., 2017). In modern Europe, the association between cold conditions and mortality in cardiovascular and respiratory diseases has weakened compared to historical times (Fonseca-Rodríguez et al., 2020, 2021). In regions characterised by warmer winters and cooler homes, a decrease in winter temperature is associated with higher increases in death rates (Eurowinter Group, 1997). In pre-industrial Europe, death from respiratory diseases showed a clear peak during the winter months, while death from gastrointestinal diseases showed a clear peak during the summer months (Buchan and Mitchell, 1875; Bradley, 1970; Lee, 1981; Wrigley and Schofield, 1981; Eckstein et al., 1984; Post, 1985; Galloway, 1987).

An additional dimension affecting seasonality in mortality is mortality displacement, or 'harvesting' (Hajat et al., 2005). This refers to that a short period of excess mortality (for example due to extreme cold or heat) could be followed by a period of lower mortality (Armstrong et al., 2017). The explanation is that mortality is displaced to occur earlier than expected among frailer individuals. A Swedish study found that high mortality rate during winter, particularly due to respiratory and cardiovascular mortality, reduced the heat effect on mortality the following summer (Rocklöv et al., 2008). This finding was supported by

90 a later Australian study, which found that the estimated heat effect on mortality was generally stronger when the preceding winter mortality was low (Qiao et al., 2015). Mortality displacement is, however, complex since they may have both short-term and long-term effect, and affect different age spans and causes of death in a different way. However, a general pattern that strong excess mortality in one season is typically followed by a reduced mortality in the next season seems to be true.

## 1.2 Malthusian demography in pre-industrial Sweden

Sweden possesses unique vital statistics at parish level back to 1749 when a 'Malthusian' demographic regime (Edvinsson, 2012), with considerable inter-annual variability in mortality in response to recurrent food crises (Larsson, 2006), still prevailed in at least large parts of the country (Dribe et al., 2017). In most other countries in Europe, in possession of early vital statistics, such a 'Malthusian' demographic regime had more or less already ended prior to the start of systematic keeping of vital statistics. This vital statistic data has, somewhat surprisingly, not hitherto been compared – across space and time – with the instrumental climate data available in Sweden back to 1722 (see, however, the pioneering study by Imhof, 1976). Such an assessment could strongly improve the understanding of climate–mortality relationship towards the end of the pre-industrial period, not only in Sweden, but for northern Europe as a whole.

During the 'Malthusian' demographic regime increased living standards tended to contribute to population growth through increased fertility and decreased mortality. In a pre-industrial 'Malthusian' society population growth was limited by so-called

positive checks (stress factors such as famines that shortened the life-span) and so-called preventive checks that decreased fertility (factors such as later marriages) (Galloway, 1988; Bengtsson et al., 2004; Klemp and Møller, 2016; Edvinsson, 2017). The interpretation of the Malthusian model, however, is diverse, with emphasis on its underlying assumptions about population growth's relationship with living standards and technological innovations. As in much of Europe, there was a sharp decline in mortality fluctuations in Sweden during the 18th century (Livi-Bacci, 2007; Edvinsson, 2017). Preventive checks on population

growth first disappeared entirely in Sweden after 1870 (Bengtsson and Dribe, 2006; Dribe, 2009). Reasonably reliable estimates of crude death and birth rates exist for Sweden (within present-day borders) back to 1630 (Edvinsson, 2015). Unfortunately, these long series of crude death rates are not separated by sex or age of the deceased. Thus, we only use the vital statistics starting in 1749, and are thus not assessing the climate–mortality relationship prior to the onset of the decline in mortality fluctuations.

Using vital statistic data from selected parishes, Larsson (2006, 2020) investigated mortality crises in Sweden during the 17th and 18th centuries. These studies have demonstrated that famines and diseases were closely intertwined during mortality crises (see also, for example, Walter and Schofield, 1989; Mokyr and Ó Gráda, 2002; Dybdahl, 2014; Ljungqvist et al., 2024). Furthermore, mortality crises during particular climate-induced shocks to food production in early modern Sweden has been investigated by Lilja (2008, 2012), and has been more comprehensively studied for the eastern part of the early modern

Swedish Realm, Finland (Huhtamaa and Helama, 2017; Huhtamaa, 2018; Huhtamaa et al., 2022). Spatial analysis using the vital statistics of mortality in certain diseases, chiefly dysentery, in present-day Sweden during the late 18th and early 19th century has been conducted by Castenbrandt (2012, 2014). The geographical distribution of malaria-attributed deaths, during peak malaria years, from 1749–1859 was assessed by Chen et al. (2021). Malaria-attributed deaths in the vital statistics

for present-day Finland has been investigated by Huldén et al. (2005) and Huldén and Huldén (2009). All these studies have revealed large inter-annual fluctuations in mortality levels.

## 1.3 Purpose and aim

The purpose of this article is to conduct a comprehensive examination of the impact of temperature and hydroclimate variability on mortality in present-day Sweden, considering both temporal and spatial dimensions. By analysing early vital statistics available from 1749 to 1859, we aim to build upon existing research and address the following key research questions: (1) How did climate variability during the study period correlate with mortality levels in different regions of Sweden? (2) What were the temporal patterns of mortality in relation to temperature variability in Sweden during the study period? (3) To what extent did temperature and hydroclimate variations contribute to mortality variations among specific subgroups (by age and sex) in Sweden during the studied period? These three research questions aim, in tandem, to provide insights into the relationship between climate factors and mortality, explore regional and temporal variations, and investigate the impact on specific population subgroups.

## 2 Materials and methods

### 2.1 Mortality data

The Tabellverket dataset is the earliest population statistics available for present-day Sweden. This dataset aggregates vital statistics from all Swedish parishes, covering the period 1749–1859, and was obtained from the Demographic Data Base (DDB) at the Centre for Demographic and Ageing Research (CEDAR) at Umeå University (Demografiska databasen, 2023). This dataset provides annual total mortality for each parish, including information on age, sex, and cause of death, albeit without specific death dates. It is important to note that the Tabellverket vital statistics have data gaps, biases, and limitations, which have been thoroughly discussed in Castenbrandt (2012). Unfortunately, prior to 1850, age is often inaccurately recorded, with discrepancies of up to 5–10 years from the actual age. This discrepancy notably introduces a bias towards an over-representation of individuals aged above 80 years (Edvinsson, 2015, p. 180). Given only three age groups considered in this study, these deficiencies are not expected to distort the result in any substantial way. In addition, there were instances of missing individuals and unrecorded deaths, though this remains a challenge even in contemporary population data.

In this study, we used all-cause death data (deaths that occur from any cause, without specific categorisation into specific disease or condition) as well as subgroups based on sex and three age groups: children (0–14), adults (15–64), and the elderly (over 65). To align with the geographical coverage of malaria-related mortality conducted by Chen et al. (2021), death data from sparsely populated districts in northern Sweden were excluded. Subsequently, the remaining part of Sweden was then divided into 49 cells, each with a grid-cell size of $1° \times 1°$ (Fig. 2). A total of 7,018,694 all-cause deaths were collected from these 49 grid-cells throughout the period 1749–1859. To ensure geographical accuracy, deaths from parishes that could not be precisely assigned to any of the 49 grid-cells were excluded from further analysis, accounting for only 2.2 % of the total

deaths (155,534 deaths out of 7,018,694 deaths) occurring between 1749 and 1859. Our study period covers 1750–1859, and information from 1749 is only included to calculate excess mortality for 1750.

      Currently there exists no monthly mortality series for Sweden for the 18th century and first part of the 19th century. Annual mortality data may not capture the seasonal nuances of mortality rates in temperate climates, where peak mortality typically occurs during the winter months (Rau et al., 2017), roughly from November to March. This limitation makes it challenging to

attribute anomalously high mortality rates during a specific year solely to the effects of exceptionally cold conditions during certain months (e.g., January–February or November–December). Despite this limitation, our study provides valuable insights into the potential connection between climate and mortality. If adverse climatic conditions have an immediate effect on mortality, we would expect correlations without time lags in our annual data. However, even in annual data, correlations with time lags may emerge if there is a delayed effect between climate conditions and mortality. Future research using more granular

monthly mortality data, feasible to obtain from selected locations, could offer a more detailed understanding of these dynamics.

## 2.2   Climate data

The longest instrumental temperature measurements within the borders of present-day Sweden started in Uppsala in 1722 (Bergström and Moberg, 2002; Moberg et al., 2002). This meteorological record is showed in Fig. 1 as it is representative of the temperature conditions in east-central Sweden and also derives from one of the most densely populated regions in

Sweden, both historically and in the present day. Other instrumental temperature measurement series than the Uppsala one started first later in the 18th century and most of them are not continuous to the present day (Brönnimann et al., 2019). For the spatial analysis, we used the monthly Berkeley Earth Surface Temperatures (BEST) gridded data with a resolution of $1° \times 1°$ since 1750 (Rohde et al., 2013a, b; Rohde and Hausfather, 2020). This gridded temperature dataset includes the Uppsala station record besides many other station records. We also used the monthly gridded Palmer Drought Severity Index (PDSI)

data on a $5° \times 5°$ grid since 1750 for studying the effects of hydroclimate (Briffa et al., 2009) (Fig. 2). This index integrates precipitation and temperature-driven evapotranspiration to estimate relative dryness (moil moisture) relative to the mean long-term conditions in a given region and tracks changes in physiological drought (Palmer, 1965; Dai et al., 2004; Wells et al., 2004; van der Schrier et al., 2011). The use of these datasets allowed us to analyse available data overlap spanning the period from 1750 to 1859.

Our study period (1750–1859) encompass the latter part of the generally cold Little Ice Age (Wanner et al., 2022). However, it is important to note that certain years and even entire decades during this period, winter (Leijonhufvud et al., 2010) as well as summer (Linderholm et al., 2015; Ljungqvist et al., 2019) temperatures in Sweden were as high as those observed in the late 20th century. Hydroclimate conditions, at least during summer, mainly fluctuated within the range of observed 20th century variations (Cook et al., 2015; Seftigen et al., 2017, 2020). The absolute temperature level, especially in spring and summer, is

uncertain prior to the late 19th century due to improper exposure of instruments among other factors, but not with regard to the amplitude of inter-annual temperature variability (Moberg et al., 2003; Böhm et al., 2010). Precipitation measurements can be considered unreliable prior to the late 19th century in Sweden (Joelsson et al., 2024), which could introduce possible biases in the PDSI data.

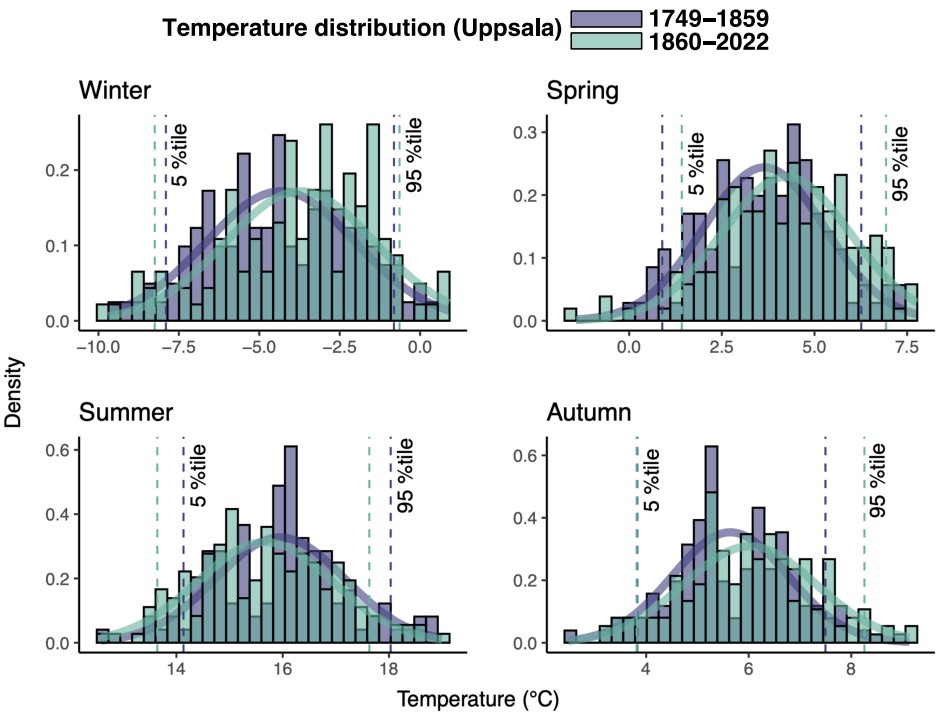

**Figure 1.** Histograms of the distribution of absolute seasonal mean temperature for winter (December–February), spring (March–May), summer (June–August), and autumn (September–November) from the Uppsala meteorological station over the study period 1749–1859 as well as for the 1860–2022 period. Shown together with the 5 % percentiles. The Uppsala temperature record is presented because it is the longest available record in present-day Sweden covering the studied period.

## 2.3 Statistical methods

In this article, it is important to differentiate between the absolute number of deaths and *excess deaths*. The term *excess deaths* refer to the deviation from the expected number of deaths within a specific period, attributed to particular events or circumstances (Rossen et al., 2020). Specifically, excess deaths were calculated as the difference between the actual number of deaths registered in a particular year $t$ and the expected number of deaths for that year $t$, estimated by a baseline derived from quasi-Poisson regression based on the entire study period 1749–1859.


$$Excess\ Deaths = Recorded\ Deaths - Expected\ Deaths \qquad (1)$$

This baseline represents a long-term trend in total deaths attributable to both the strong population growth and declining mortality rates throughout this period (Fig. 3a). The time-series of mortality were then correlated with monthly climate observations.

Additionally, another commonly used method, also tested here, involved deriving a baseline from a linear regression using

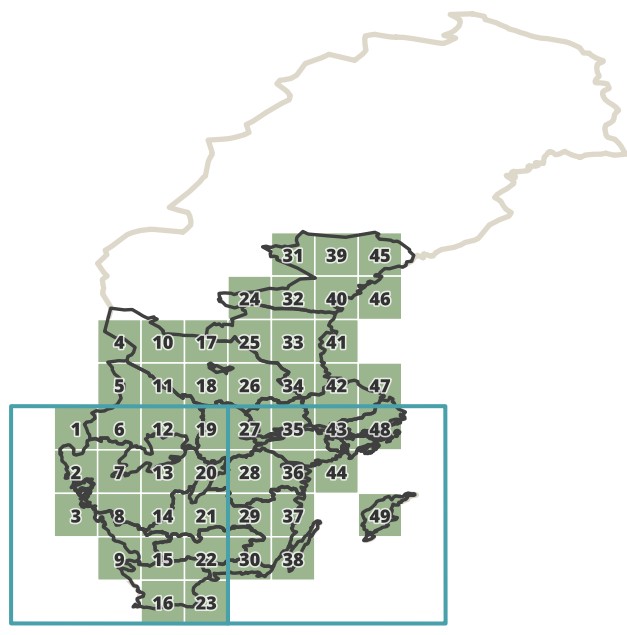

**Figure 2.** The 49 different 1°×1° grid-cells (in green) included for the spatial analysis between climate data and mortality in Sweden. Two 5°×5° grid-cells (in blue) are used for the Palmer Drought Severity Index (PDSI) data.

mortality or seasonal data from the preceding five years (see Fig. 3b). To facilitate comparisons across regions with different population sizes and age structures, we further employed the *P-score* metric (Msemburi et al., 2023). The *P-score* represents the percentage difference between the actual and expected number of deaths. For instance, if the expected number of deaths for a particular year *t* is 100 and the actual number of deaths is 120, the excess deaths would be 20 and the *P-score* would be

20 %, indicating that the death count is 20 % higher than expected for year *t*. This metric allows for a standardised measure and is calculated as follows:

$$Excess\ Mortality\ (P\text{-}score) = \frac{Recorded\ Deaths - Expected\ Deaths}{Expected\ Deaths} * 100 \qquad (2)$$

While we consider the *P-score* of excess mortality to be the most pertinent metric for this study, it has its limitations. For

instance, in periods of heightened mortality, it may not reflect an increase when evaluated in relation to a trend deviation. Additionally, excess mortality do not differentiate between for example war-related deaths, certainly not directly tied to climate, and other deaths more likely to be associated with climate like malnutrition-driven diseases. In addition to this metric, we also used national-level crude death rate data (Statistics Sweden, 1999; Edvinsson, 2015) for comparison with various baselines results (Fig. 3c). Given that crude death rate data already accounts for population change, we applied the quasi-Poisson regression

baseline to remove the decreasing trend in mortality.

To investigate potential temporal variations in the relationship between climate and mortality, we also conducted experiments by dividing the mortality data into two separate periods: an earlier period spanning from 1750 to 1804, and a later period from 1805 to 1859. For each of these periods, we employed the quasi-Poisson regression baseline to calculate excess mortality. The significance levels in this article refer to $p = 0.05$ with a two-tailed Student's $t$-test. All data integration, spatial analyses, and mapping were performed using a combination of the software **FME** (Safe Software Inc., 2023), **QGIS** (QGIS, 2023), and **R** (R Core Team, 2022).

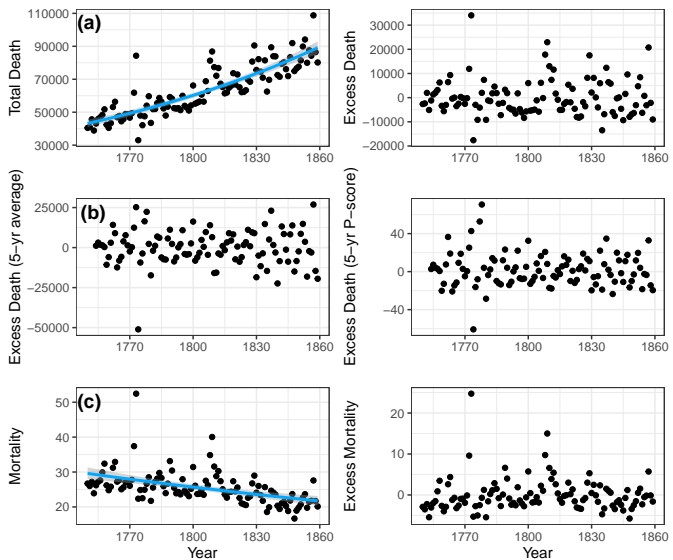

**Figure 3.** Comparison of baselines and estimates of excess mortality. (a) *Left:* Total number of deaths within 49 grid-cells and quasi-Poisson baseline (blue line); *Right:* Estimates of excess mortality. (b) *Left:* Excess death using five-year average baseline; *Right: P*-score using five-year average baseline. (c) *Left:* Crude death rate and quasi-Poisson baseline (blue line); *Right:* Estimates of excess mortality. Source: the period 1630–1759 from Edvinsson (2017) and the period 1760–1860 from Statistics Sweden (1999).

## 3   Results

### 3.1   Climate–mortality relationships on a national level

We found a statistically significant negative relationship between late winter and spring temperature – in particular February–April temperature – and mortality for the entire studied population both for the same and for the following year (Table 1; Fig. 4). Colder late winters and springs were associated with higher mortality and *vice versa*. The spread of the data around the trend line indicates some variability in the relationship, but the overall pattern remains consistent, reflecting the statistically significant negative correlation. Despite a correlation coefficient of around –0.3, may seem relatively weak, it is still rather noteworthy that about 10 % of the fluctuations in mortality can be explained by late winter–spring temperature variations,

given the many different causes of mortality. The effect of summer and autumn temperature on mortality was non-significant
– regardless whether the temperature effect on mortality for the same or the following year is considered. Similar results were
also found when using the Uppsala temperature station data, instead of the gridded Berkeley temperature dataset, to correlate
against national-level Swedish mortality data (Table A1). In general, the Uppsala temperature data showed higher correlations
with mortality compared to the Berkeley temperature dataset, which may indicate a higher reliability of the Uppsala temperature
series.

In our analysis of climate–mortality relationships categorised by sex (Table 1) and age groups (Table 3), we observed almost
entirely identical mortality patterns between women and men ($r = 0.99$ over the 1749–1859 period). Among the three age
groups examined, the most statistically significant associations were found for the 15–64 age group, with correlations reaching
up $r = –0.39$ between April temperature of the same year and mortality. The mortality pattern for the 15–64 age group differed
from that of the over-65 age group. In the former, we observed a statistically significant negative correlation between spring
temperature and mortality, both during the same year and the following year. On the other hand, lower late winter and spring
temperatures increased mortality among the over-65 age group only during the same year. Conversely, for the 0–14 age group,
we found an increased mortality during the following year after a year with cold late winter and spring conditions.

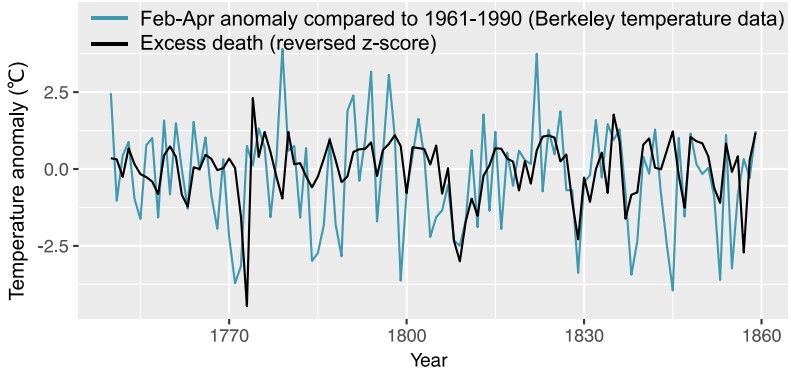

**Figure 4.** The negative correlation ($r = –0.28$) found between February–April mean temperature (shown here with regard to the 1961–1990
mean) using the Berkeley temperature data and standardised excess mortality of the same year (z-score shown in opposite sign).

## 3.2  Geographical patterns of the climate–mortality relationships

We investigated the geographical pattern of the temperature–mortality associations for the late winter–spring season found
to exhibit the strongest such relationship. Regardless the length of seasonal windows, we consistently observed the strongest
correlations in two regions: southern-most Sweden for the impact of temperature on mortality during the same year, and east-
central Sweden (specifically the Svealand region) for the effect of temperature on mortality in the following year (Fig. 5).
Time-series of the excess mortality for each of the 49 grid-cells is shown in Fig. A2. Maps showing the correlation coefficients

**Table 1.** Correlation coefficient ($R$) between the total excess mortality (estimated by quasi-Poisson regression) and the Berkeley temperature data (the average of all 49 grid-cells) over the entire 1750–1859 period. Values which reached statistical significance ($p < 0.05$) using a two-tailed Student's $t$-test are marked in bold.

| Month(s) | *R* same year<br>All persons | *R* following year<br>All persons | *R* same year<br>Men (all ages) | *R* following year<br>Men (all ages) | *R* same year<br>Women (all ages) | *R* following year<br>Women (all ages) |
|---|---|---|---|---|---|---|
| January | –0.07 | –0.09 | –0.07 | –0.08 | –0.06 | –0.09 |
| February | –0.13 | **–0.29** | –0.13 | **–0.28** | –0.12 | **–0.29** |
| March | **–0.26** | **–0.19** | **–0.26** | **–0.19** | **–0.25** | **–0.19** |
| April | **–0.32** | **–0.26** | **–0.33** | **–0.26** | **–0.32** | **–0.26** |
| May | 0.02 | –0.10 | 0.02 | –0.09 | 0.02 | –0.11 |
| June | –0.03 | –0.09 | –0.02 | 0.09 | –0.03 | 0.09 |
| July | 0.06 | 0.00 | 0.06 | –0.01 | 0.05 | 0.01 |
| August | 0.17 | –0.14 | 0.17 | –0.16 | 0.17 | –0.13 |
| September | 0.05 | –0.14 | 0.05 | –0.14 | 0.04 | –0.14 |
| October | 0.05 | –0.07 | 0.06 | –0.07 | 0.05 | –0.07 |
| November | 0.14 | 0.01 | 0.15 | 0.00 | 0.13 | 0.01 |
| December | 0.12 | 0.05 | 0.14 | 0.05 | 0.11 | 0.05 |
| DJF | –0.07 | **–0.22** | –0.08 | **–0.21** | –0.07 | **–0.23** |
| FMA | **–0.28** | **–0.32** | **–0.29** | **–0.31** | **–0.28** | **–0.32** |
| MA | **–0.33** | **–0.26** | **–0.34** | **–0.26** | **–0.33** | **–0.26** |
| MAM | **–0.29** | **–0.26** | **–0.29** | **–0.26** | **–0.28** | **–0.27** |
| AMJ | **–0.19** | –0.16 | **–0.19** | –0.15 | **–0.19** | –0.17 |
| JJA | 0.02 | 0.04 | 0.03 | 0.04 | 0.01 | 0.05 |
| SON | 0.07 | –0.13 | 0.07 | –0.13 | 0.06 | –0.13 |
| DJFMAM | **–0.19** | **–0.29** | **–0.20** | **–0.27** | –0.18 | **–0.29** |
| Annual mean | –0.07 | **–0.25** | –0.07 | **–0.25** | –0.08 | **–0.25** |

**Table 2.** Correlation coefficient ($R$) between the total excess mortality (estimated by *P-score* using five-year average baseline) and the Berkeley temperature data (the average of all 49 grid-cells) over the entire 1754–1859 period. Values which reached statistical significance ($p < 0.05$) using a two-tailed Student's *t*-test are marked in bold.

| Month(s) | *R* same year All persons | *R* following year All persons | *R* same year Men (all ages) | *R* following year Men (all ages) | *R* same year Women (all ages) | *R* following year Women (all ages) |
|---|---|---|---|---|---|---|
| January | 0.04 | –0.04 | 0.03 | –0.04 | 0.05 | –0.03 |
| February | –0.10 | –0.15 | –0.10 | –0.14 | –0.10 | –0.16 |
| March | **–0.25** | –0.02 | **–0.26** | –0.02 | **–0.25** | –0.02 |
| April | –0.09 | 0.01 | –0.10 | 0.02 | –0.08 | 0.01 |
| May | 0.09 | –0.11 | 0.09 | –0.10 | 0.09 | –0.12 |
| June | –0.04 | 0.10 | –0.04 | 0.11 | –0.04 | 0.10 |
| July | –0.01 | –0.08 | 0.00 | –0.09 | –0.01 | –0.07 |
| August | 0.11 | –0.16 | 0.11 | –0.17 | 0.11 | –0.15 |
| September | 0.07 | –0.06 | 0.07 | –0.06 | 0.06 | –0.06 |
| October | –0.11 | –0.06 | –0.11 | –0.07 | –0.12 | –0.07 |
| November | **–0.31** | 0.15 | **–0.31** | 0.13 | **–0.31** | 0.16 |
| December | **–0.25** | 0.06 | **–0.26** | 0.05 | **–0.24** | 0.07 |
| DJF | 0.11 | –0.02 | 0.11 | –0.01 | 0.11 | –0.02 |
| FMA | **–0.20** | –0.08 | **–0.20** | –0.08 | **–0.19** | –0.09 |
| MA | **–0.22** | –0.01 | **–0.23** | –0.01 | **–0.22** | –0.01 |
| MAM | –0.16 | –0.05 | –0.17 | –0.04 | –0.16 | –0.05 |
| AMJ | –0.03 | 0.00 | –0.03 | 0.01 | –0.02 | –0.01 |
| JJA | –0.03 | 0.01 | –0.02 | 0.00 | –0.03 | 0.01 |
| SON | –0.05 | –0.08 | –0.04 | –0.08 | –0.06 | –0.08 |
| DJFMAM | 0.12 | –0.08 | 0.12 | –0.09 | 0.11 | –0.07 |
| Annual mean | 0.03 | –0.08 | 0.03 | –0.08 | 0.03 | –0.07 |

**Table 3.** Correlation coefficient ($R$) between the total excess mortality (estimated by quasi-Poisson regression) for age-specific groups and the Berkeley temperature data (the average of all 49 grid-cells) over the entire 1750–1859 period. Values which reached statistical significance ($p < 0.05$) using a two-tailed Student's $t$-test are marked in bold. $n$ = sample size (in millions).

| Month(s) | *R* same year Ages 0–14 $n = 240$ [41 %] | *R* following year Ages 0–14 | *R* same year Ages 15–64 $n = 194$ [33 %] | *R* following year Ages 15–64 | *R* same year Ages 65+ $n = 151$ [26 %] | *R* following year Ages 65+ |
|---|---|---|---|---|---|---|
| January | 0.02 | –0.08 | –0.13 | –0.06 | –0.13 | –0.01 |
| February | 0.02 | **–0.31** | –0.14 | **–0.21** | **–0.25** | –0.18 |
| March | –0.18 | **–0.21** | **–0.25** | **–0.24** | **–0.27** | –0.08 |
| April | –0.13 | –0.17 | **–0.39** | **–0.36** | **–0.28** | –0.14 |
| May | 0.10 | –0.14 | 0.03 | –0.01 | –0.12 | –0.02 |
| June | –0.03 | 0.02 | –0.01 | 0.14 | –0.11 | 0.10 |
| July | 0.10 | 0.00 | 0.07 | 0.08 | –0.06 | 0.08 |
| August | 0.19 | –0.14 | 0.16 | –0.01 | 0.00 | –0.08 |
| September | 0.14 | –0.10 | 0.06 | –0.02 | –0.13 | –0.15 |
| October | 0.08 | –0.06 | 0.06 | 0.02 | 0.02 | 0.03 |
| November | 0.17 | 0.01 | 0.06 | 0.02 | 0.07 | 0.10 |
| December | 0.13 | 0.07 | 0.08 | 0.01 | –0.02 | –0.08 |
| DJF | 0.03 | **–0.23** | –0.13 | –0.14 | **–0.22** | –0.15 |
| FMA | –0.13 | **–0.31** | **–0.31** | **–0.33** | **–0.34** | –0.17 |
| MA | –0.19 | **–0.23** | **–0.36** | **–0.34** | **--0.32** | –0.12 |
| MAM | –0.13 | **–0.25** | **–0.31** | **–0.30** | **–0.33** | –0.12 |
| AMJ | –0.04 | –0.16 | **–0.21** | –0.14 | **–0.27** | –0.05 |
| JJA | 0.05 | 0.01 | 0.04 | 0.13 | –0.09 | 0.10 |
| SON | 0.14 | –0.10 | 0.08 | 0.01 | –0.05 | –0.06 |
| DJFMAM | –0.04 | **–0.29** | **–0.24** | **–0.25** | **–0.32** | –0.17 |
| Annual mean | 0.08 | **–0.24** | –0.13 | –0.17 | **–0.28** | –0.11 |

with other seasonal windows with the most significant temperature–mortality relationship – namely February, March, April, March–April, and April–June – are shown in Figs. A4–A8.

The relationship between hydroclimate (soil moisture) and mortality is much weaker than the temperature–mortality associations. In terms of regional variations, the association between hydroclimate and mortality during the same year appear stronger in wetter western Sweden compared to drier eastern Sweden (Table A2). Generally, wetter conditions – especially in winter and

spring – during the same year were associated with weakly increased mortality. However, this association reached statistical significance solely for January and April and only in western Sweden. Regarding the effect on mortality for the following year, wetter conditions one year – especially during summer and autumn – tended to increase mortality in both western and eastern Sweden during the following year. This relationship was statistically significant for August–October in eastern Sweden, and also for September and December in western Sweden (Table A2).

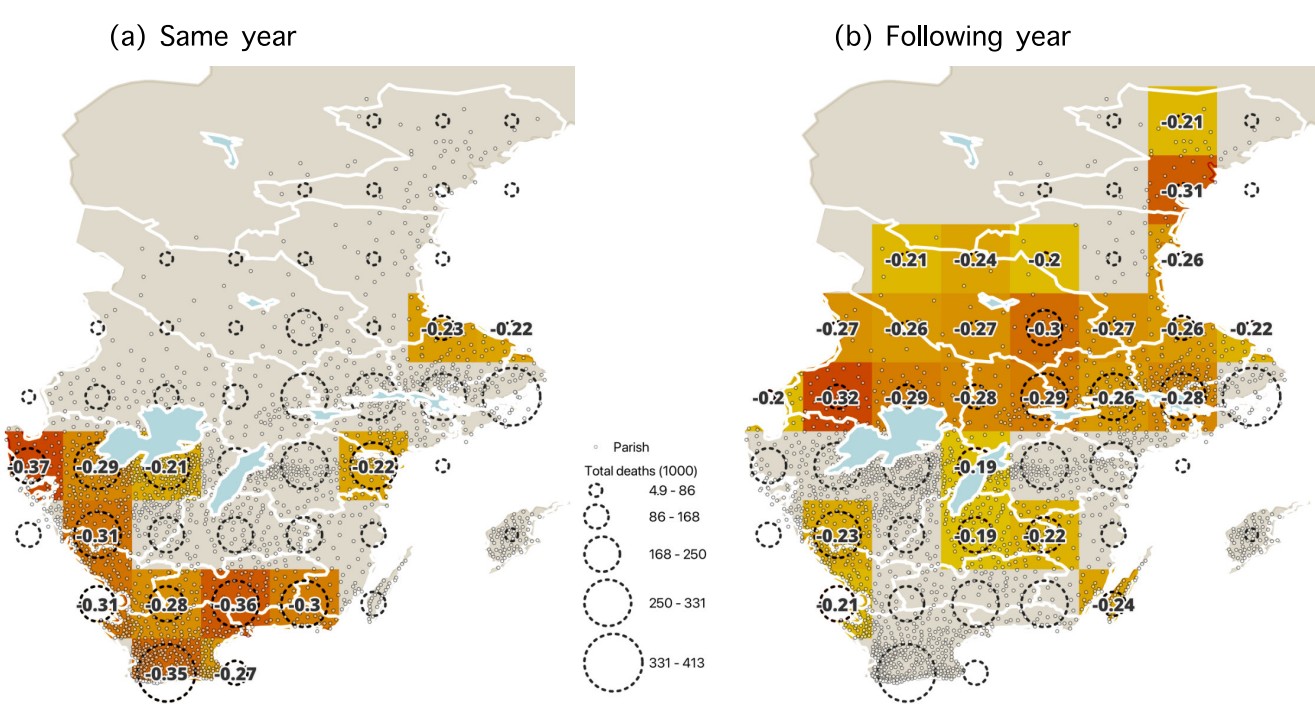

**Figure 5.** Statistically significant correlation coefficients ($p < 0.05$) during the 1750–1859 period between February–April temperature and mortality for (a) the same year, and (b) the following year. Stronger correlations are indicated by darker colours on the maps. Parishes are represented by dots, and dashed circles represent the number of total deaths (1000) from parishes within each grid-cell.

## 3.3 Changes in the climate–mortality relationships over time

Using a 31-year centered moving window correlation of February–April mean temperature and total annual mortality in all the 49 grid-cells from Sweden (excess death estimated through quasi-Poisson regression), we observed distinct patterns in the mortality response during different time periods (Fig. 6). In the 1790s, a statistically significant increase in the mortality response for the same year is detected, with a running correlation exceeding $r = -0.36$ (thus explaining about 13 % of the mortality variation). This strong correlation persisted throughout most of the following period, gradually weakening after approximately 1830. The effect of February–April temperature on the mortality the following year remained statistically significant above $r = -0.36$ until 1824. However, this correlation rapidly weakened and became non-significant towards the end of the study period.

To further investigate the effect of different months on mortality, we divided the study period 1750–1859 into an earlier period (1750–1804) and a later period (1805–1859) (Table 4). The temperature effect on the mortality during the same year was much stronger during the later period (1805–1859) than during the earlier period (1750–1804). The correlation reached $r = -0.53$ for April and $r = -0.49$ for March–April during the 1805–1859 period. This means that about 25 % of the variation in mortality is explained. Conversely, the temperature effect on the following-year mortality was stronger during the earlier period for most monthly and seasonal windows, and decreased from $r = -0.45$ to $-0.23$ from the earlier to later period. Moreover, we found that higher November and December temperatures resulted in significantly higher mortality during the 1750–1804 period, but this effect was not observed during the later period. In addition, a statistically significant increase in mortality in response to higher August temperature during the same year was only observed at all during the 1805–1859 period. In summary, the temperature effect on the mortality during the same year was stronger during the latter period (as shown in Fig. 6). Interestingly, the temperature effect on the mortality during the same year in Scania (southern-most part of Sweden) weakened over time and shifted to eastern Svealand (around Stockholm). However, for the temperature effect on mortality the following year, this effect was observed in large regions, but gradually became insignificant over time (Fig. 7).

## 3.4 Extreme cold-related and warm-related mortality

We further analysed the geographical patterns of the relationship between February–April temperature and mortality for the same year and the following year by especially focusing on the six coldest (< 5th percentile) and six warmest (> 95th percentile) February–April seasons between 1750 and 1859 (Fig. 8). The coldest February–March seasons during this period occurred in 1845, 1771, 1799, 1853, 1838, and 1829, while the warmest such seasons were observed in 1779, 1822, 1794, 1797, 1750, and 1791 (also shown in Fig. 4). The results reveal a diffuse pattern when compared to the long-term relationship shown in Fig. 5. In the southern-most region of Sweden, Scania, where the long-term February–April temperature had the strongest long-term effect on the mortality during the same year, this effect was not particularly pronounced during neither the extreme cold or warm years. However, for the effect on mortality during the following year, some extremely cold February–April seasons (1772, 1800, 1839) exhibited similar patterns in central Sweden (Fig. 8a) as the persistent temperature–mortality relationship. Similarly, some extremely warm February–April season (1823, 1798, 1751) were linked to lower mortality (Fig. 8b). Unexpectedly, the warmest February–April season, 1779, was followed by higher mortality.

**Figure 6.** Moving correlation using a 31-year centered moving window with 95 % confidence interval (shaded areas) between February–April mean temperature and mortality (excess mortality estimated by quasi-Poisson regression). Dashed lines indicate statistical significance level at $p < 0.05$ and $p < 0.01$.

## 4 Discussion

### 4.1 Climate–mortality relationships: time-lags, geographical patterns, and subgroups

We have observed geographical differences in the winter and spring temperature effect on mortality between the far south-western part of Sweden and central Sweden. The temperature effect on mortality during the same year was much stronger in the former region, whereas the temperature effect on mortality in the following year was much stronger in the latter region (Section 3.2). Galloway (1994) found a statistically significant mortality increase in response to colder winters in Sweden, and to a lesser extent to colder springs, whereas we found a stronger spring temperature effect than winter temperature effect on

mortality. However, we did not detect a statistically significant mortality increase in response to warmer summers, contrary to the findings of Galloway (1994). These discrepancies may be attributed to methodological differences, as we considered correlation coefficient between time-series, while Galloway (1994) considered lag sum responses. Similarly to our study, Eckstein et al. (1984) found little influence of summer temperatures on pre-industrial Swedish mortality. However, Schumann et al. (2013) reported increased mortality in Uppsala due to higher summer temperatures. This could presumably be explained by the

fact that the Uppsala region experiences higher summers temperatures, particularly higher maximum temperatures, compared to most of Sweden (Wastenson et al., 1995). While our findings also demonstrate a small influence of hydroclimatic conditions on mortality in late pre-industrial Sweden (Section 3.2), it is important to note that this phenomena is not only weak but also exhibits geographical and temporal variations. The hydroclimatic signal is so weak, albeit partly statistically significant, that

# Mortality response to February-April temperature

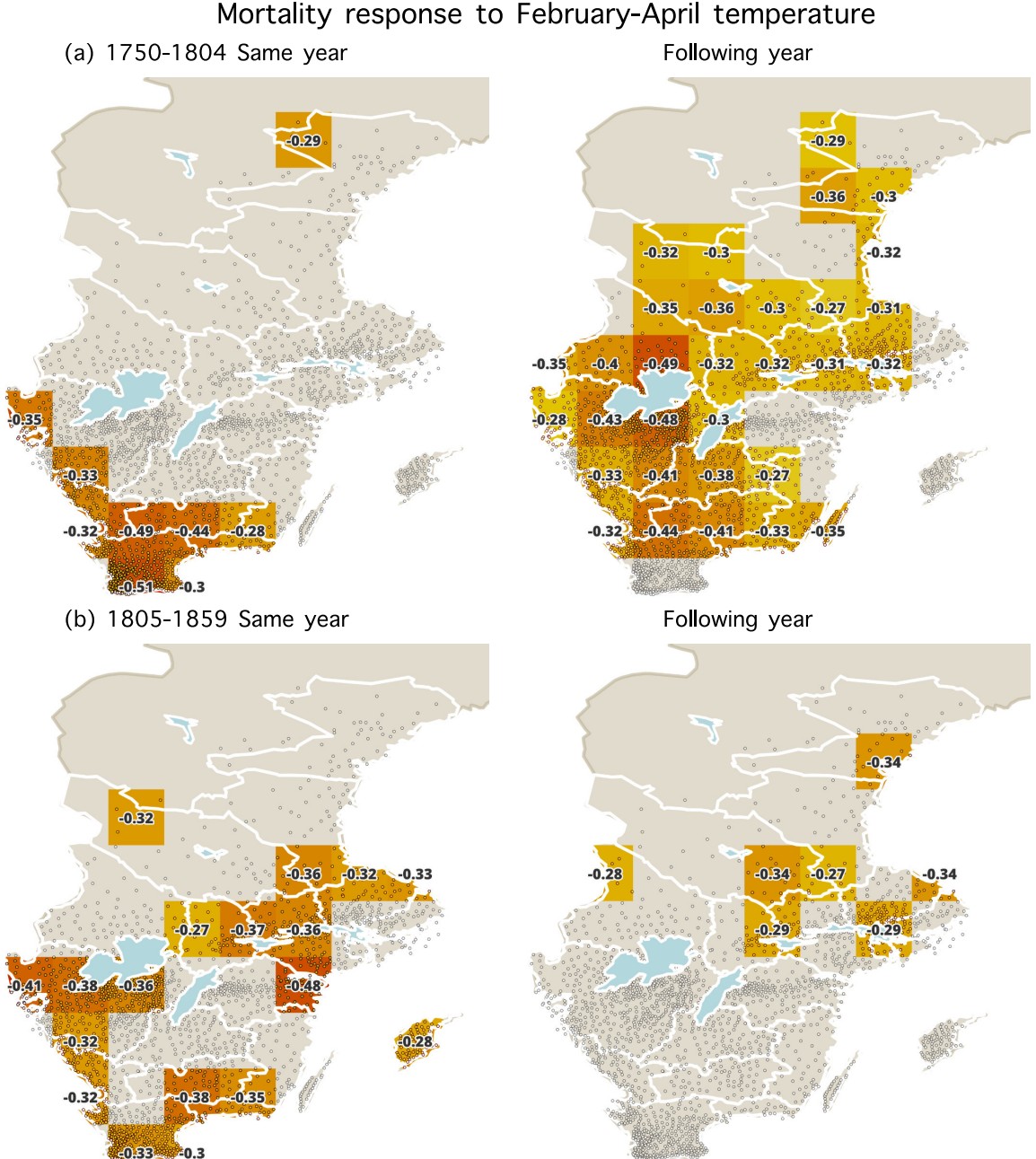

(a) 1750-1804 Same year

Following year

(b) 1805-1859 Same year

Following year

**Figure 7.** Statistically significant correlation coefficients ($p < 0.05$) between February–April average temperature and mortality (all persons) during the same year and the following year for the periods (a) 1750–1804 and (b) 1805–1859, respectively. Stronger correlations are indicated by darker colours on the maps. Parishes are represented by dots.

**Table 4.** Correlation coefficient ($R$) between mortality (all persons) and the Berkeley temperature data for periods 1750–1804 and 1805–1859. Values which reached statistical significance ($p < 0.05$) using a two-tailed Student's $t$-test are marked in bold.

| Month(s) | R same year 1750–1804 | R following year 1750–1804 | R same year 1805–1859 | R following year 1805–1859 |
|---|---|---|---|---|
| January | 0.10 | –0.11 | –0.18 | –0.06 |
| February | –0.09 | **–0.43** | –0.17 | **–0.22** |
| March | **–0.23** | **–0.34** | **–0.35** | –0.09 |
| April | –0.01 | **–0.21** | **–0.53** | **–0.22** |
| May | 0.06 | –0.19 | 0.04 | 0.03 |
| June | –0.16 | 0.11 | 0.14 | 0.14 |
| July | –0.07 | –0.04 | 0.20 | 0.08 |
| August | 0.04 | **–0.28** | **0.29** | –0.04 |
| September | 0.03 | –0.13 | 0.11 | –0.12 |
| October | 0.06 | –0.12 | 0.13 | 0.07 |
| November | **0.36** | **–0.26** | –0.07 | **–0.28** |
| December | **0.25** | 0.13 | 0.00 | –0.06 |
| DJF | 0.01 | –0.02 | 0.05 | –0.15 |
| FMA | –0.16 | **–0.45** | **–0.40** | **–0.23** |
| MA | –0.18 | **–0.35** | **–0.49** | –0.17 |
| MAM | –0.14 | **–0.37** | **–0.40** | –0.13 |
| AMJ | –0.05 | –0.17 | **–0.21** | –0.05 |
| JJA | –0.13 | 0.03 | 0.19 | 0.12 |
| SON | 0.06 | –0.15 | 0.18 | –0.01 |
| DJFMAM | 0.13 | –0.02 | –0.18 | **–0.37** |
| Annual mean | 0.06 | **–0.30** | –0.14 | –0.17 |

we leave it out of the discussion. However, we emphasise the importance to recognise that multiple climatic factors contribute to variations in mortality rates in the late pre-industrial era in Sweden, in particular during specific individual extreme years.

Although the absence of a temperature effect on the following year's mortality in southwestern-most Sweden can be explained by the region's agriculture being less sensitive to a late onset of the growing season (Skoglund, 2022, 2024), one could have expected central Sweden to also display an effect of cold winters and springs on the same year's mortality. The increased mortality observed the year following colder late winters and springs in central Sweden can almost certainly be attributed to reduced food supply resulting from adverse climatic effects on agriculture (Edvinsson et al., 2009; Dribe et al., 2017). However, this should not, in principle, preclude an impact of temperature on mortality during the same year as well. It is highly plausible that during the 18th century, when the mortality rate was generally higher, the impact of cold winters and springs on increased

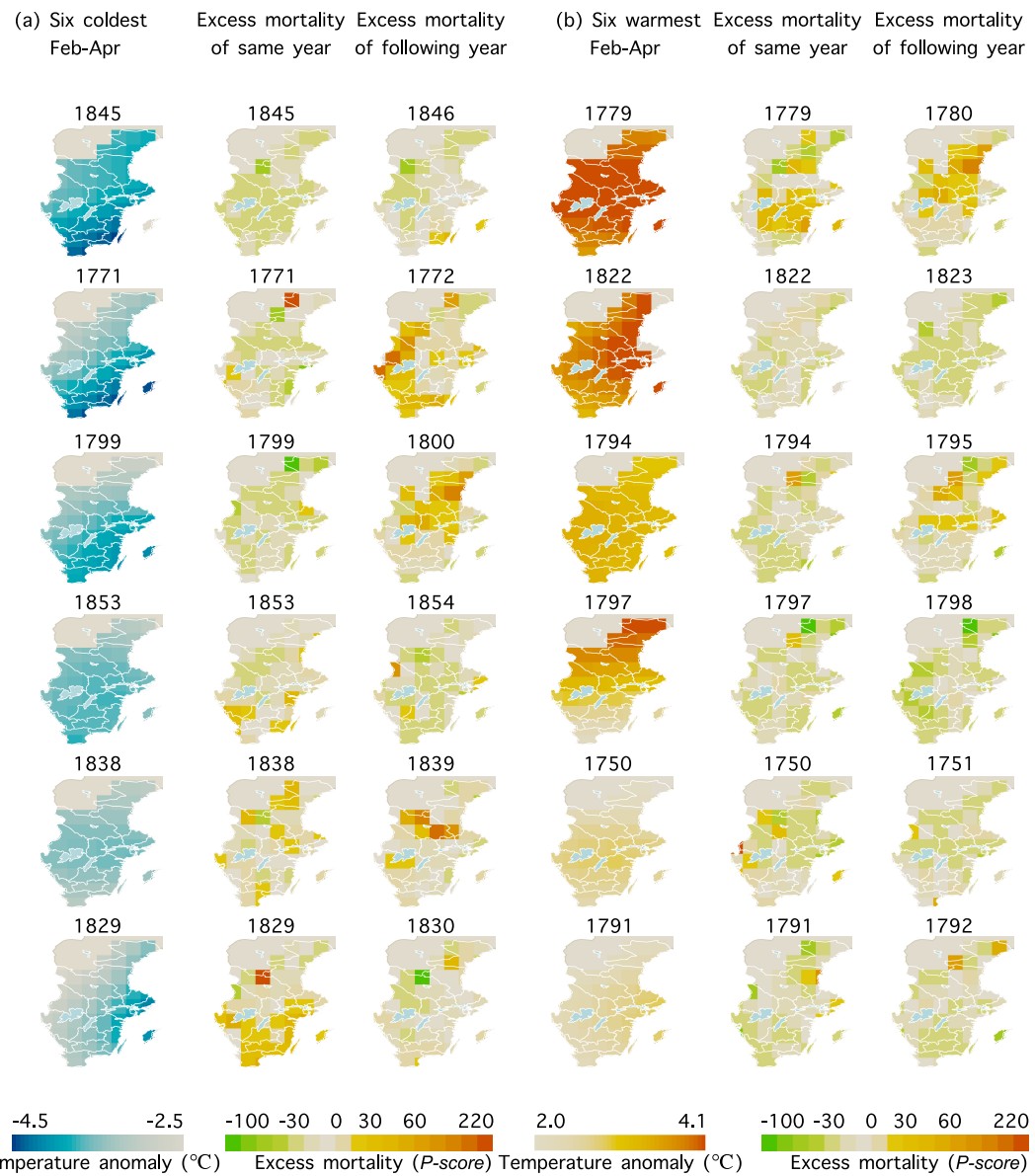

**Figure 8.** Spatial patterns of mortality (here we used *P-score* in order to compare the relative increase in excess mortality across regions with different population sizes) on the (a) six coldest (<5th percentile) and (b) six warmest (>95th percentile) February–April seasons from 1750 to 1859, for the same and the following year. The years are arranged in descending order from the coldest/warmest (*top*) to the least cold or warm (*bottom*). Temperature anomaly values are relative to the 1961–1990 mean derived from Berkeley temperature data.

mortality was less discernible compared to the 19th century, when the mortality rate had decreased. With a lower mortality rate, it can be expected that even smaller increases in mortality in response to extreme climatic conditions would appear more

prominently amidst the overall larger variability in mortality. One likely explanation for the weakening relationship after about 1825 is the improved living conditions, including better nutrition, during the 19th century. Overall mortality, mainly from infectious diseases, decreased, and fluctuations in death rates were reduced. For example, smallpox was nearly eradicated with the mandatory vaccination introduced in 1816 (Sköld, 1996).

Whereas men and women were equally affected by changes in temperature, the impact of temperature on mortality differed between age groups. Among the 0–14 age group, mortality increased during the following year after a year with colder spring and winter conditions (Section 3). This can probably be largely attributed to the effects of spring temperatures on food production (Edvinsson et al., 2009; Holopainen et al., 2012) and, thus, on nutrition level (Ljungqvist et al., 2024). It is likely that this especially affected infant mortality, which accounted for much of the mortality within the 0–14 age group in 18th and 19th century Sweden (Statistics Sweden, 1969). For the age group 15–64, the negative association between late winter and spring temperature and mortality can be attributed to several factors. Cold winter and spring conditions during the pre-industrial era have been associated with the increased spread of respiratory and contagious diseases (Galloway, 1994). This is likely due to indoor crowding and decreased immune system resistance to respiratory infections in cold temperature (Eurowinter Group, 1997). Such respiratory deaths occur rapidly and contribute to higher mortality within the same year. In addition, the increased mortality during the year following cold late winters and springs can be explained by adverse effects on agriculture triggering increased malnutrition. Moreover, a noteworthy negative association was observed between the same year's winter and spring temperature and mortality among the elderly (65 and older). Cold temperatures could have directly impacted the health of older individuals, making them more susceptible to respiratory infections, cardiovascular events, and other cold-related health risks (Fonseca-Rodríguez et al., 2020). For example, influenza mortality tends to increase with cold and dry conditions (Lowen et al., 2007; Lowen and Steel, 2014). An area to further investigate, in future research, is the potential mortality displacement ('harvesting effect') among the elderly during cold years with higher mortality that could then be presumed to be followed by lower mortality and weaker effects of exposure the following year(s).

## 4.2   The effects of climate-induced harvest variations on mortality

Late winter and early spring temperatures have been found to have played a critical role for grain production in much of the main agricultural districts in south-central Sweden (Edvinsson et al., 2009; Holopainen et al., 2012). Warm winters and springs tended to increase harvests as long as late spring and summer droughts did not suppress them (for the latter, see Ljungqvist et al., 2023). Furthermore, cold late winters and springs had an adverse effect on diary production in Sweden (Utterström, 1955) as it had in regions with comparatively milder climate such as the British Isles (Costello et al., 2023) and Germany (Baten, 2001). Thus, it is likely that all, or most, of the negative relationship between late winter and spring temperature and the following year's mortality was due to climate effects on food availability. In fact, the relationship between temperature and the mortality of the following year, strongest in central Sweden and prior to the 1820s, can hardly be explained in another way than through changes in food availability. Its geographical pattern, being largely restricted to the Svealand region, needs to be further investigated both in terms of the climate sensitivity of the agriculture in the region and the societal resilience to food shortage in this region compared to in southern-most Sweden. The mortality associated with malnutrition in the 18th and

19th centuries was primarily driven by disease outbreaks (Mokyr and Ó Gráda, 2002; Larsson, 2006, 2020; Ljungqvist et al., 2024). However, to better understand the extent to which climate-related mortality fluctuations can be attributed to starvation or malnutrition, further research is needed to investigate the causes of death recorded in the vital statistics from that period.

The following-year mortality response to climate variability is thus presumed to mainly operate through affecting the nutritional situation through the intermediary of harvest outcome and, to a lesser extent, livestock mortality. We acknowledge that the exclusion of harvest data, as a central intermediate stage between climate and mortality, constitute a major limitation in the present study. However, we leave the complex issue of harvest–mortality relationships to further studies due to several issues compromising the reliability and representativeness of 18th and 19th century Swedish grain harvest data. First, yield ratio series were typically collected from individual farms, which may not accurately reflect entire regions and may not even be locally representative, as they tended to be biased towards the most prosperous farms or manors (Slicher van Bath, 1963). Second, tithes are frequently used as an indicator of harvest volume (Leijonhufvud, 2001); however, tithes in Sweden captured only 25 % to 50 % of the actual harvests, with some being fixed amounts. During the study period, only small gradually decreasing portions of Sweden still paid taxes in tithes reflecting the actual harvest variations (Hallberg et al., 2016). Third, harvest data do not necessarily translate into actual food availability, as grain and other food sources could be, and were, imported from other regions (Edvinsson, 2012) or from aboard (Åmark, 1915). The third point, however, is of limited relevance here as the observed association between climatic and following-year mortality must reasonably be interpreted as a result of local to regional harvest outcomes.

At present, a reconstruction of Swedish harvests (Edvinsson, 2009) is available for our study period only until 1820 and, as a national average, has limited explanatory value for studying regional harvest–mortality relationships. This harvest reconstruction, however, indicates an increasing per capita harvest from about 1810, and with setbacks already by the 1790s, signifying increased food security. The per capita harvests, when including potatoes, were about 15–20 % higher around 1820 than they had been about thirty years earlier (Edvinsson, 2009). This agrees well with our finding that the relationship between winter–spring temperature and the following year's mortality became non-significant during the 1820s. Furthermore, we note that this concurs with the findings by Larsson (2006). In his work with mortality data from selected parishes, Larsson (2006) also interpreted increased mortality during the winter–spring season as indicative of malnutrition due to previous year harvest failures, and found that these seasonal mortality peaks declined over time as a result on an improved food supply situation.

## 4.3 Methodological considerations

When estimating excess mortality, or any change in mortality beyond normal fluctuations that is attributed to an external factor, the choice of comparison period or baseline is crucial (Section 2.3). This baseline represents a reference for estimating the expected mortality level in the absence of the variable of interest. While the concept of excess mortality has been studied extensively, the COVID-19 pandemic has further highlighted the importance of this methodology. The challenge lies not only in properly estimating the baseline, but also in finding a method that allows for meaningful comparisons between regions or countries. This complexity arises from the fact that different countries, or regions, may have experienced varying trends in mortality and population dynamics. Different methodologies have been developed and evaluated, ranging from simple strate-

gies such as comparing death counts from the previous year, to advanced regression models that aim to account for changes in both mortality levels and population characteristics.

In this study, we tested two different methods for estimating baseline, or expected, mortality: a baseline fitted from a quasi-Poisson regression baseline, and a baseline derived from a linear regression based on the mortality during the past five years. The use of a five-year average, or trend, as a baseline for studying excess mortality during the COVID-19 pandemic was a common and straightforward approach (Barnard et al., 2022; Nepomuceno et al., 2022; Wang et al., 2022; Levitt et al., 2023). It can be effective when mortality rates are relatively stable over time. However, in our study period characterised by significant

fluctuations in mortality, where sudden events like wars or epidemics can have a substantial but temporary impact on annual mortality, the five-year baseline becomes a sensitive reference period. An outlier event can significantly influence the baseline, leading to either an overestimation or an underestimation of excess mortality. Moreover, this method does not account for any long-term trends in mortality, which can be a limitation. This issue is more pronounced in modern times when mortality generally shows improvement with every consecutive year. In our study period, the year-to-year variability in mortality posed

the main challenge. To address this issue, we adopted a reference period that encompassed the full study period, using the quasi-Poisson regression model. This approach allowed us to account for the long-term trends and mitigate the impact of inter-annual fluctuations in mortality. As a sensitivity analysis, we also stratified the time period into two segments to explore any potential temporal variations in the climate–mortality relationship. Interestingly, the results from this stratified analysis aligned closely with those obtained from the full study period, confirming the robustness of our findings.

As a result, the excess mortality using quasi-Poisson regression baseline (Table 1) demonstrated stronger correlations with temperature changes compared to the five-year approach (Table 2), aligning with findings from other studies comparing different baseline estimation methods. While the methods may yield different levels of excess, the overall patterns tend to remain consistent (Nepomuceno et al., 2022; Levitt et al., 2023). The *P*-score, however, proved to be a more suitable measure for comparing the relative increase in excess mortality across regions with different population sizes, and possibly different age

structures. This is due to the inherent tendency for larger population sizes and older age groups to have higher excess death counts. In addition, the comparison of excess crude death rates and excess deaths using quasi-Poisson regression baseline (Table A3) revealed similar outcomes. Both measures showed statistically significant correlations with temperature, suggesting that excess death using quasi-Poisson regression baseline can effectively serve as a measure for excess mortality.

There are several limitations to consider in this study. First, the use of annual resolved mortality data restricted our possibility

for examining the seasonality of climate-related mortality, which would have provided valuable insights of climate–mortality relationship. Second, our analysis primarily focused on the individual effects of temperature, and to a limited extent, hydroclimate variables, and not using classified climate types, such as hot/wet, hot/dry, cold/wet, or cold/dry. An inclusive consideration of climate variables would provide an effective assessment of the risks associated with different diseases. Moving forward, future research could gain a deeper understanding by exploring the climate–mortality relationship in different time periods,

geographical regions, and in particular, cause-specific mortality. In addition, the integration of more comprehensive dataset, including factors such as land-use and socio-economic indicators, would provide a more nuanced understanding of region-specific mortality patterns and their relationship with climate. Incorporating data on grain harvests and regional grain import

could enable a more thorough examination of the extent to which food availability were mediated through climate impacts. Furthermore, a full understanding of the causes of climate-related mortality is only possible by studying changes in the cause of deaths in relation to various climate parameters. The data from Tabellverket would, in principle, allow for investigating this. However, it would require a major research undertaking.

## 5    Conclusions

Our study has revealed a statistical significant association between temperature and mortality in Sweden during the period 1750–1859. Colder temperatures in late winter and spring were found to elevate mortality rates during the same year, with this impact extending also into the following year. This effect of February, March, and April temperature were particularly large both for mortality during the same and the following year. Exploring geographical differences in the temperature–mortality relationship, we found that the southern-most regions experienced the greatest impact of temperature on the mortality during the same year. Conversely, central Sweden exhibited the strongest temperature effect on mortality in the following year. This suggests that in central and northern Sweden, the influence of temperature on mortality was primarily attributed to the adverse effects of cold conditions on food production, while this was not the case in southern Sweden. To establish causation definitively, future studies should incorporate data on both regional harvests and monthly mortality.

Child mortality was notably influenced by effect of cold late winter and spring conditions the previous year, likely associated with malnutrition. In contrast, mortality among the elderly was more immediate and potentially linked to cold-related diseases that affected health within the same year. It is noteworthy that the temperature–mortality relationship changed over time, showing a diminishing impact after the 1820s, especially for following-year mortality. This indicates improved food and nutrition security during that period. Further research is needed to refine and expand upon these findings. This includes a more in-depth exploration of the underlying factors that contributed to the variation in the climate–mortality relationship patterns between the southern and central regions of Sweden, as well as a detailed analysis of the specific diseases and other causes of death associated with season- and age-related mortality.

*Code availability.*    We have used **FME** (Safe Software Inc., 2023), **QGIS** (QGIS, 2023), and **R** (R Core Team, 2022) to program the analysis codes used in this work. The **R** code and **FME** workflow are available upon reasonable request.

*Data availability.*    The entire population mortality data can be accessed through the open data SHiPS platform (http://ships.ddb.umu.se/) at the Centre for Demographic and Ageing Research (CEDAR), Umeå University, Sweden. However, the age-specific mortality data used in the current article are not publicly available and require a license. These data can be made available upon reasonable request and with the permission of CEDAR. The Berkeley Earth Surface Temperatures (BEST) is licensed under Creative Commons BY-NC 4.0 International for non-commercial use only and is available from the website https://berkeleyearth.org/data/. The Uppsala temperature station data (Bergström and Moberg, 2002) is freely available from the Swedish Meteorological and Hydrological Institute: https://www.smhi.se/data/meteorologi/temperatur/uppsalas-

temperaturserie-1.2855. The PDSI data from Briffa et al. (2009) is freely available from Climatic Research Unit at the University of East Anglia: https://crudata.uea.ac.uk/cru/data/drought/europe.

*Author contributions.* T.T.C. helped to conceive and design the study, collected and integrated the data, conducted the statistical and geostatistical analyses, created figures, and wrote part of the text. R.E. provided expertise on pre-industrial demography in Sweden. K.M. provided expertise regarding the calculation of excess mortality and on modern climate–mortality relationships. H.W.L provided valuable input on study design and provided climatological expertise. F.C.L. conceived and designed the study and wrote much of the text.

*Competing interests.* Hans W. Linderholm is a member of the editorial board of *Climate of the Past*.

*Acknowledgements.* F.C.L. acknowledges a Visiting Researcher stay at the Freiburg Institute for Advanced Studies (FRIAS), that allowed him time to finalise the work with this article. The authors thank the two anonymous reviewers, and Bertil Forsberg at Umeå University, whose useful comments helped improve this article.

*Financial support.* T.T.C. was supported by the Swedish Research Council Formas (grant no. 2017-01161), and by the Bolin Centre for Climate Research, Stockholm University (grant no. RT3. BC: 30002632). H.W.L. was funded by the Swedish Research Council (grant no. 465 2021-03886). F.C.L. was supported by the project "Adapting to climate change in the northern Baltic Sea region, AD 1500–1900" funded by Marianne and Marcus Wallenberg Foundation (grant no. MMW 2022-0114), and he conducted the work with this article as a Pro Futura Scientia XIII Fellow funded by the Swedish Collegium for Advanced Study through Riksbankens Jubileumsfond. Open access publication funding for this article was provided by Stockholm University.

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

## Appendix A

Table A1. Correlations of excess mortality (estimated by quasi-Poisson regression) against Uppsala temperature station data (Moberg and Bergström, 1997) over the period 1749–1859. Values which reached statistical significance ($p < 0.05$) using a two-tailed Student's $t$-test are marked in bold.

| Month(s) | *R* same year All persons | *R* following year All persons | *R* same year Men | *R* following year Men | *R* same year Women | *R* following year Women |
|---|---|---|---|---|---|---|
| January | –0.05 | –0.09 | –0.05 | –0.08 | –0.05 | –0.10 |
| February | –0.18 | **–0.39** | –0.18 | **–0.38** | –0.17 | **–0.40** |
| March | **–0.28** | **–0.22** | **–0.28** | **–0.22** | **–0.27** | **–0.22** |
| April | **–0.32** | **–0.27** | **–0.32** | **–0.27** | **–0.31** | **–0.27** |
| May | 0.01 | –0.09 | 0.01 | –0.08 | 0.02 | –0.10 |
| June | –0.08 | 0.08 | –0.08 | 0.08 | –0.09 | 0.08 |
| July | 0.04 | 0.01 | 0.04 | –0.01 | 0.04 | 0.03 |
| August | 0.11 | –0.15 | 0.11 | –0.16 | 0.11 | –0.14 |
| September | –0.02 | –0.18 | –0.02 | –0.18 | –0.02 | **–0.19** |
| October | 0.04 | –0.09 | 0.04 | –0.09 | 0.03 | –0.09 |
| November | 0.16 | 0.03 | 0.16 | 0.02 | 0.15 | 0.03 |
| December | 0.11 | 0.02 | 0.12 | 0.02 | 0.10 | 0.02 |
| DJF | –0.10 | **–0.27** | –0.11 | **–0.26** | –0.10 | **–0.28** |
| FMA | **–0.32** | **–0.39** | **–0.32** | **–0.38** | **–0.31** | **–0.39** |
| MA | **–0.35** | **–0.29** | **–0.36** | **–0.29** | **–0.35** | **–0.29** |
| MAM | **–0.30** | **–0.28** | **–0.30** | **–0.28** | **–0.29** | **–0.29** |
| AMJ | **–0.20** | –0.16 | **–0.20** | –0.15 | **–0.19** | –0.16 |
| JJA | 0.024 | –0.03 | 0.04 | –0.04 | 0.03 | –0.02 |
| SON | 0.11 | –0.10 | 0.12 | –0.10 | 0.10 | –0.10 |
| DJFMAM | **–0.22** | **–0.33** | **–0.22** | **–0.32** | **–0.21** | **–0.34** |
| Annual mean | –0.13 | **–0.31** | –0.13 | **–0.31** | –0.13 | **–0.31** |

**Table A2.** Correlations of excess mortality (estimated by quasi-Poisson regression) with PDSI data over the entire period 1750–1859. Sweden is divided following longitude 15°E into a western and an eastern part. Values which reached statistical significance ($p < 0.05$) using a two-tailed Student's $t$-test are marked in bold.

| Month | *R* same year Western Sweden | *R* following year Western Sweden | *R* same year Eastern Sweden | *R* following year Eastern Sweden |
|---|---|---|---|---|
| January | **0.21** | 0.10 | 0.03 | –0.06 |
| February | 0.19 | 0.12 | 0.04 | –0.03 |
| March | 0.16 | 0.11 | 0.02 | –0.03 |
| April | **0.22** | 0.19 | 0.07 | 0.02 |
| May | 0.18 | 0.17 | 0.06 | 0.03 |
| June | 0.15 | 0.16 | 0.09 | 0.05 |
| July | 0.09 | 0.19 | 0.06 | 0.03 |
| August | 0.09 | 0.19 | 0.04 | 0.02 |
| September | 0.08 | **0.20** | 0.05 | 0.03 |
| October | 0.05 | 0.18 | 0.01 | 0.00 |
| November | 0.02 | 0.19 | –0.02 | –0.02 |
| December | 0.02 | **0.21** | –0.01 | 0.01 |

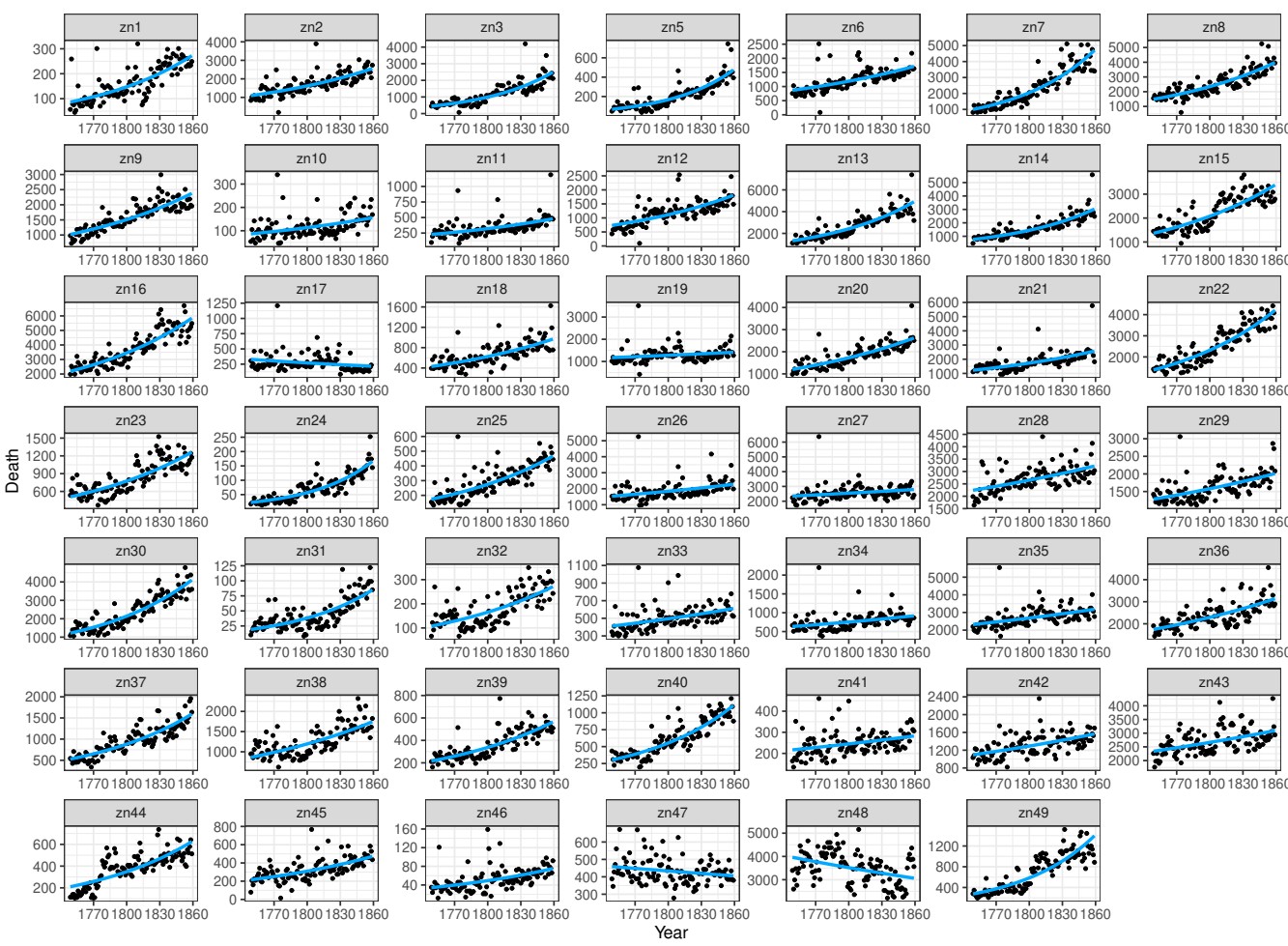

**Figure A1.** Respective baselines estimated by quasi-Poisson regression. No parish was located within grid-cell no. 4, hence no data presented.

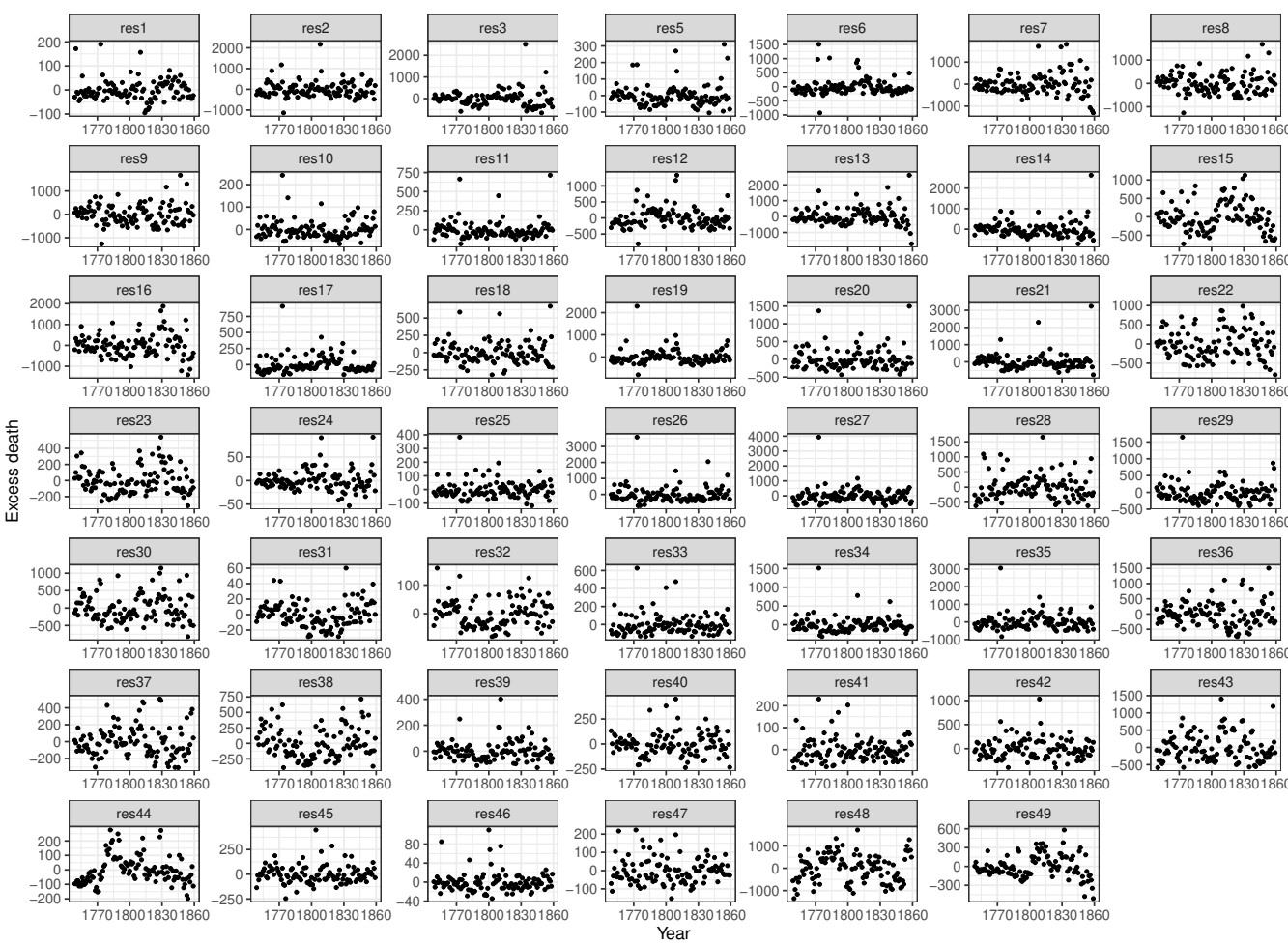

**Figure A2.** Excess all-cause death using quasi-Poisson baseline for all age groups of each 49 grid-cell during 1749–1859. Excess deaths were estimated by the difference between actual deaths and the baseline estimated for each grid-cell.

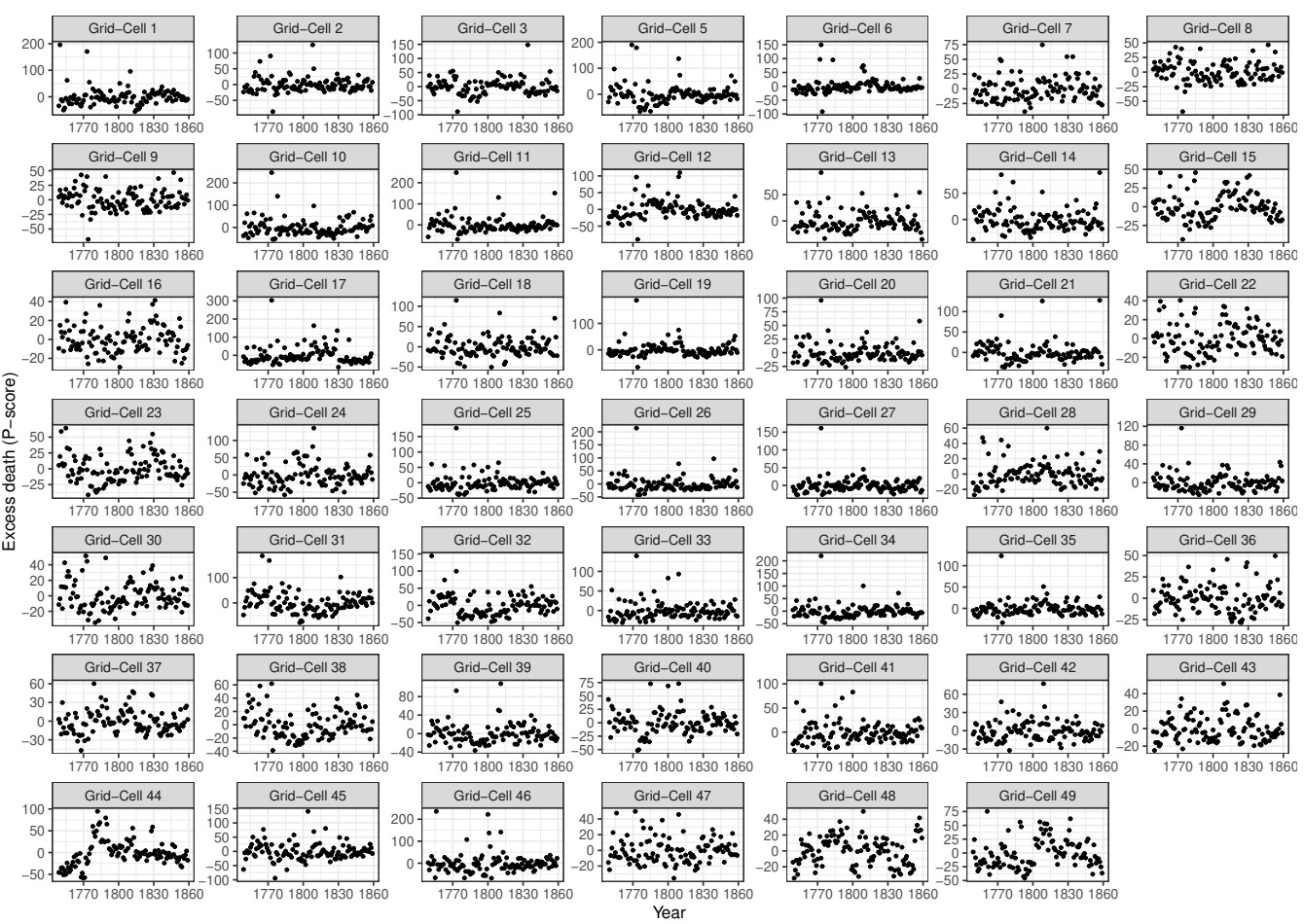

**Figure A3.** Excess all-cause death as *P*-score using quasi-Poisson baseline for all age groups of each 49 grid-cell during 1749–1859. No parish was located within grid-cell no. 4, hence no data is presented.

**Table A3.** Comparison of correlations of excess mortality (estimated by quasi-Poisson regression) and excess crude death rate for all persons with the Berkeley temperature data from 49 grid-cells during 1750–1859. Values which reached statistical significance ($p < 0.05$) using a two-tailed Student's $t$-test are marked in bold.

| Month(s) | *R* same year Excess death | *R* following year Excess death | *R* same year Excess crude death rate | *R* following year Excess crude death rate |
|---|---|---|---|---|
| January | –0.07 | –0.09 | – 0.04 | -0.05 |
| February | –0.13 | **–0.29** | -0.11 | **–0.28** |
| March | **–0.26** | **–0.19** | **–0.27** | **–0.31** |
| April | **–0.32** | **–0.26** | **–0.25** | **–0.27** |
| May | 0.02 | –0.10 | 0.09 | –0.07 |
| June | –0.03 | –0.09 | –0.04 | 0.11 |
| July | 0.06 | 0.00 | 0.06 | 0.06 |
| August | 0.17 | –0.14 | 0.15 | –0.07 |
| September | 0.05 | –0.14 | 0.07 | –0.02 |
| October | 0.05 | –0.07 | 0.09 | 0.01 |
| November | 0.14 | 0.01 | 0.15 | 0.11 |
| December | 0.12 | 0.05 | 0.13 | 0.03 |
| DJF | 0.03 | –0.09 | –0.03 | –0.10 |
| FMA | **–0.28** | **–0.32** | **–0.26** | **–0.37** |
| MA | **–0.33** | **–0.26** | **–0.31** | **–0.34** |
| MAM | **–0.29** | **–0.26** | **–0.24** | **–0.33** |
| AMJ | **–0.19** | –0.16 | –0.12 | –0.14 |
| JJA | 0.02 | 0.04 | 0.02 | 0.09 |
| SON | 0.07 | –0.13 | 0.11 | 0.00 |
| DJFMAM | **–0.19** | **–0.28** | 0.02 | –0.15 |
| Annual mean | –0.07 | **–0.25** | –0.03 | **–0.20** |

## Mortality response to February temperature

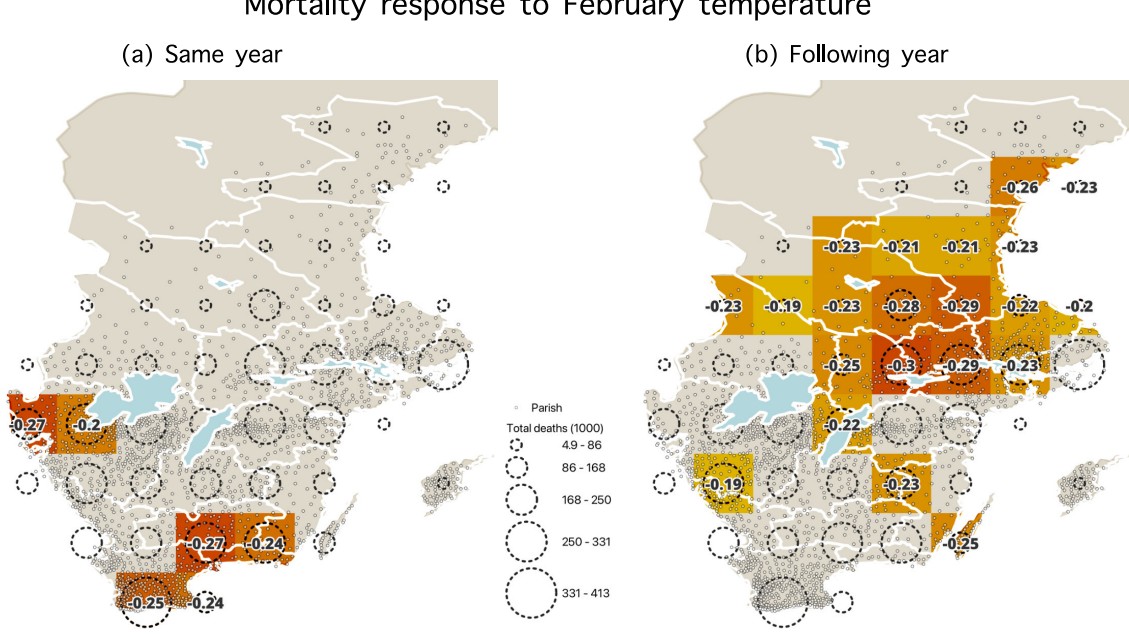

**Figure A4.** Statistically significant correlations ($p < 0.05$) during the 1750–1859 period between February temperature and mortality for (a) the same year, and (b) the following year. Parishes are shown in dots, and dashed circles represent the number of total deaths (1000) from parishes within each grid-cell.

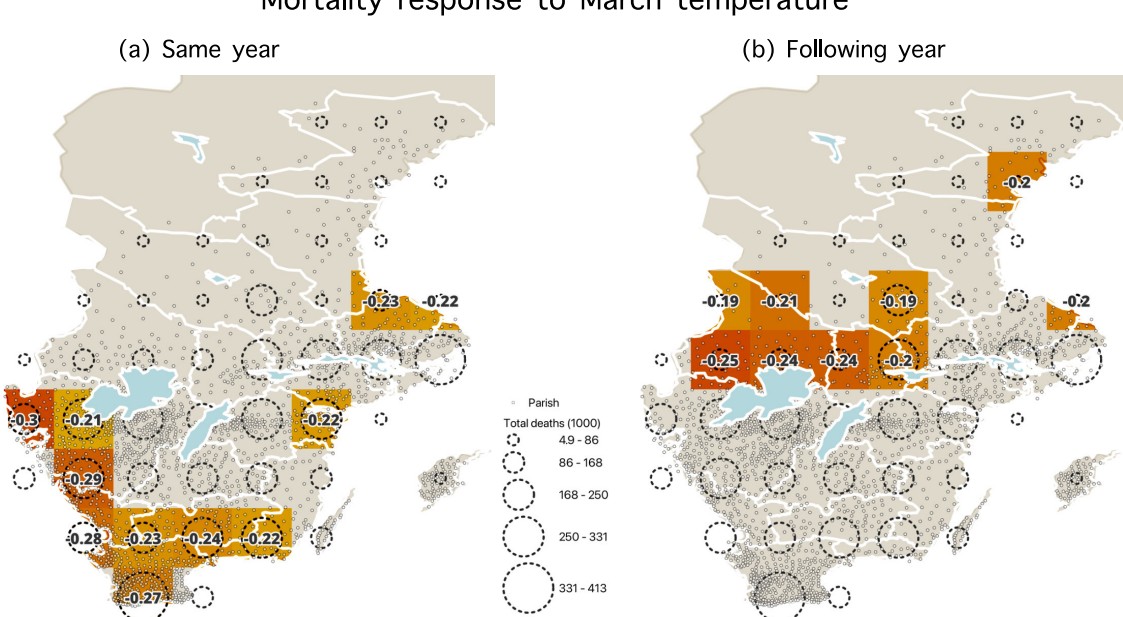

**Figure A5.** Statistically significant correlations ($p < 0.05$) during the 1750–1859 period between March temperature period and mortality for (a) the same year, and (b) the following year. Parishes are shown in dots, and dashed circles represent the number of total deaths (1000) from parishes within each grid-cell.

## Mortality response to April temperature

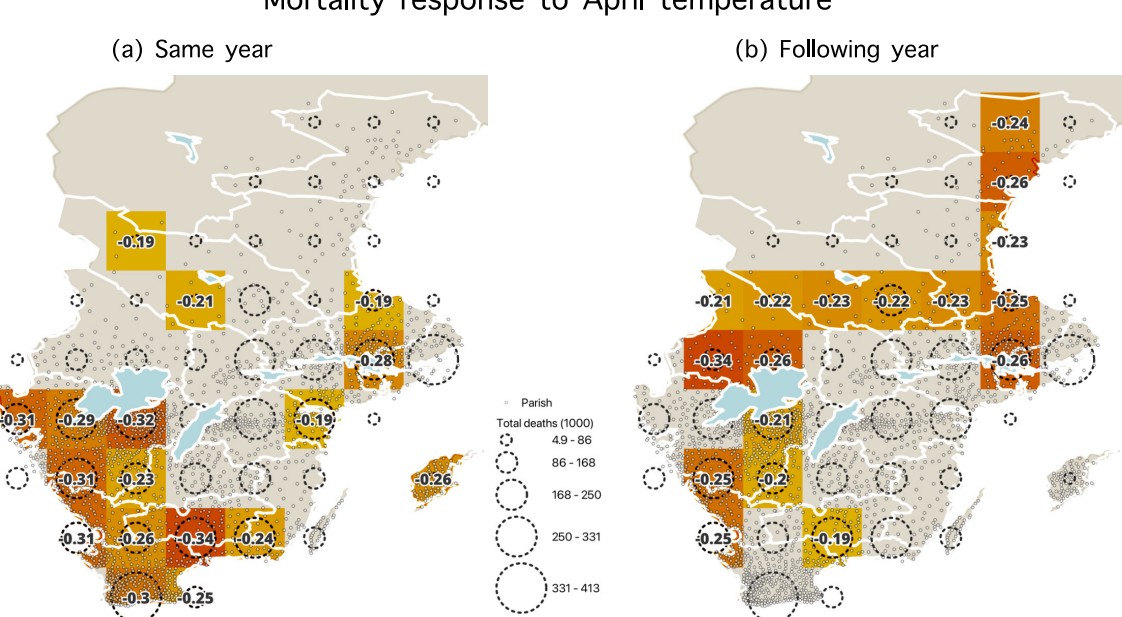

**Figure A6.** Statistically significant correlations ($p < 0.05$) during the 1750–1859 period between April temperature and mortality for (a) the same year, and (b) the following year. Parishes are shown in dots, and dashed circles represent the number of total deaths (1000) from parishes within each grid-cell.

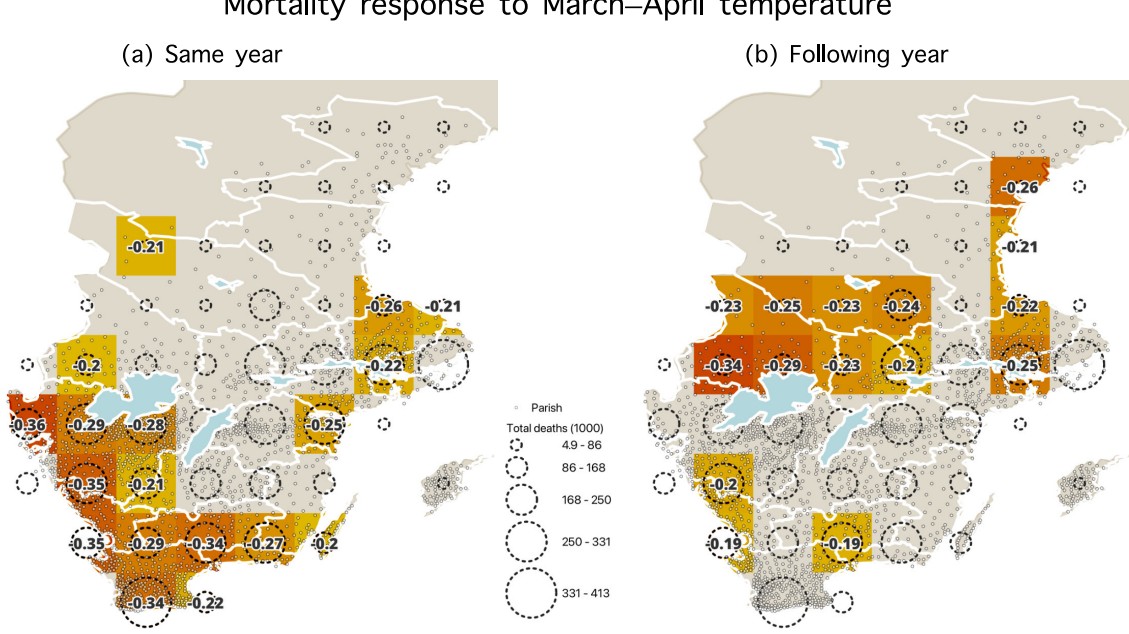

**Figure A7.** Statistically significant correlations ($p < 0.05$) during the 1750–1859 period between March–April temperature and mortality for (a) the same year, and (b) the following year. Parishes are shown in dots, and dashed circles represent the number of total deaths (1000) from parishes within each grid-cell.

## Mortality response to April–June temperature

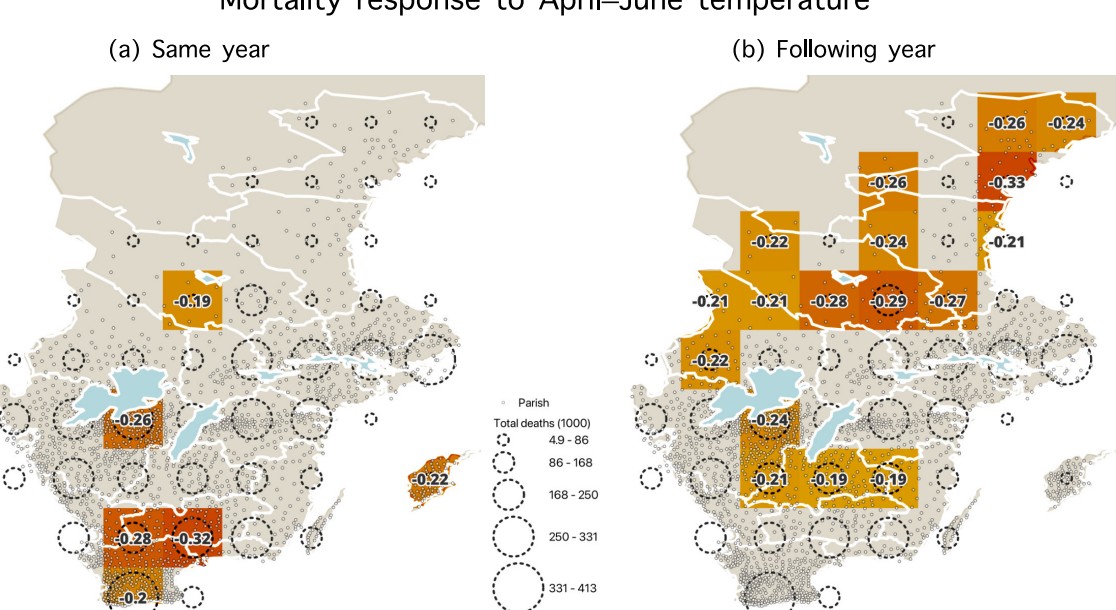

**Figure A8.** Statistically significant correlations ($p < 0.05$) during the 1750–1859 period between between April–June temperature and mortality for (a) the same year, and (b) the following year. Parishes are shown in dots, and dashed circles represent the number of total deaths (1000) from parishes within each grid-cell.