# Peer review of "Climatic impacts on mortality in pre-industrial Sweden"

_Climate of the Past, 2023_

## Referee Comment (RC1)

Climate impacts on mortality in pre-industrial Sweden

T.T. Chen, R- Edvinsson, K. Modig, H.W. Linderholm, F. C Ljungqvist

Recommendation: To be published after some changes.

1.- Introduction

It includes an excellent review of the literature. I suggest to add some information from the public health point of view, including a description of the annual cycle of mortality rate whose maximum should coincide with the coldest months of the year. I found an article that claims that for cohorts born in 1800 the risk of dying during the winter season was almost twice that of dying during summer (Ledberg, 2020)

2.- Material and methods

I suggest to indicate the sizes of the three age groups: 0-14 years, 15-64 years, over 65 years.

It is not clear the purpose of including Fig. 1 showing the seasonal histograms of the Uppsala temperature record. As this information is not used later on in the analysis, I suggest to eliminate it.

In line 145 the study period is defined as 1750 – 1859.  However in other parts of the text is mentioned as 1749 - 1859 (lines 6, 95, 111, 135, 138, 158, figure caption of Fig. 1)

It is a general consensus that in temperate climates mortality rate is highest during winter, let say from November to March. Then, as the data for mortality rate is only available on an annual basis, the causes for an anomalously high value of mortality for a certain year can be attributed to anomalously cold weather either at the beginning of the year (January – February), at the end of the year (November – December) or both at the beginning or at the end of the year. This difficulty for the interpretation of the results is not mentioned when presenting the available data.

3. Results

The correlation technique is generally used to test an hypothesis for a relationship between two variables. What it would be the hypothesis for a relationship between the mean regional temperature for a single month and the annual rate of mortality that are behind the correlations presented in Tables 1, 2, 3, 4, A1 and A2 ?  Is it plausible to consider that mean temperature during a single month will have an impact on the annual rate of mortality ? This question if particularly valid for months at the end of the year, considering that to a large extent the mortality rate is determined by the deaths that occurred during the previous months.

I suggest to use a more precise language when extracting conclusions from correlations relatively low in magnitude, although statistically significant. For example, from correlations presented in Table 1 it is mentioned in line 190 "Colder winters and springs were associated  with higher mortality and vice versa". In fact, the correlations of the order of -0.3 explain only 9% of the mortality variance. A scatter diagram (which is not presented) would illustrate the weakness of the

link among the two variables. Later on in the text it is shown that the relatively low magnitude of the correlations derives from the fact that the relationship between temperature and mortality rate during winter and spring does not persist during the entire study period, due to geographical changes in its intensity (see Fig. 7).

I suggest to eliminate the discussion about the relationship between the rate of mortality and the PDSI index. Correlations presented in Table A2 are mostly not statistically significant and explain at most 4% of the variance of mortality. Furthermore, it is mentioned in line 151 – 152 that precipitation measurements can be considered unreliable prior to the late 19$^{th}$ century in Sweden. Regarding this, in the discussion section there is another example of a statement that overrates the results that were obtained. Line 307: "While our findings demonstrate the influence of a wet autumn on mortality in both western and eastern Sweden…". This statement is not supported by the results presented in Table A2, showing very low and mostly not statistically significant correlations between mortality and PDSI data for March, April and May.  As in the case of temperature, it is possible that the positive correlations are stronger for certain periods, but this analysis was not performed.

4.- Discussion

There are several references to results showing the impact of winter and spring temperature on mortality (i.e. lines 255, 270, 282, 293). However, surprisingly, most of the correlations shown in Tables 1 – 4, A1 and A2 for December, January and February are not statistically significant.  It is more correct to indicate that the results reveal some impact of low temperature on mortality during late winter and spring. Same objection is valid for statements in the Abstract and in the Conclusions.

5.- Conclusions

Lines 367 – 369: "… we found that the southern -most regions experienced the greatest impact of temperature on mortality during the same year. Conversely, central Sweden exhibited the strongest temperature effect on mortality in the following year"

Comment: In my opinion this conclusion does not summarize in an adequate way the results presented in Fig. 7. This figure shows significant geographical changes of these impacts during the study period. Thus, in the eastern – central part of the country anomalously low temperature during FMA tended to be associated with above average mortality rate in the period 1805-1859, but not in the period before. Do the author have and hypothesis for this ?.  On the other hand, while during the period 1750 – 1804 relatively large mortality rate tended to prevail in the whole region the year following an anomalously cold FMA, this relationship was restricted to the central-eastern portion of the country afterward. Do the authors consider that improved food and nutrition security did not reach that part of the country  during that period (1804 – 1859) ?

Lines 371 . 372: "Among adults, colder conditions in April had the most adverse effect on mortality both for the same year and the following year"

Comments: I question this statement referring to the impact of temperature during a single month, considering that Table 3 shows correlations similar in magnitude to that in April for the periods February – April, March – April and March April and May.

Reference

Ledberg, A., 2020. A large decrease in the magnitude of seasonal fluctuactions in mortality explains part of the increase in longevity in Sweden during the 20[th] century.

---

## Author Response (AR1)

**Editor Dr. Chantal Camenisch, *Climate of the Past***

Fredrik Charpentier Ljungqvist
Professor of History, especially Historical Geography

Department of History
Stockholm University
SE-106 91 Stockholm
Sweden

E-mail: fredrik.c.l@historia.su.se
Mobile phone: +46706620728

[Figure]

Monday, September 09, 2024

**Manuscript revision**

Dear editor Dr. Chantal Camenisch,

Thank you very much for the reviews of our manuscript "Climatic impacts on mortality in pre-industrial Sweden" and for the opportunity to revise the manuscript. We again thank all the reviewer for their comments and suggestions. As you wrote in your editorial decision, we already "have answered the comments and questions in their replies very satisfactorily", and we have thus not included a point-by-point response. Regarding the three unaddressed comments, that you mentioned, we have now addressed them too in the following ways:

(1) We now explain more the inclusion of the Uppsala temperature record in the text.

(2) We have now rephrased the parts about the relationship between mortality and the PDSI index.

(3) We have decided not to include an actual scatter-plot but have instead, in the text, written more about the nature of the relationship between temperature and mortality. In case you as editor strongly insists that we must include scatter-plots, we can of course include them, but after discussions within the author team we found that they would bring little added value in light of how we now have tuned and rephrased the text.

Only substantial changes and additions are marked in red in the track-change version. Corrections of minor typos and changes of individual words are not marked (it is difficult using LaTex).

We hope that the revised version of the manuscript is now suitable for publication in *Climate of the Past*. Thank you very much and we are looking forward to hearing from you soon.

On behalf of all authors,

Fredrik Charpentier Ljungqvist

---

## Referee Report (RR1)

I appreciate the depth with which you have addressed each point of my comments, especially in expanding on the climate-mortality relationship and explaining the challenges with integrating historical harvest data. I understand your decision to leave this aspect for future studies, as the variability and potential inaccuracies of the data, particularly for tithe series, as you say, present challenges. Given the period studied here, using annual yield reports – despite their own limitations – might provide a useful input in future analyses. While I understand that this is outside the scope of the current study, exploring harvest data in future research could shed light on the specific impact of weather on mortality, independent of harvest outcomes.

I also see the logic in focusing on correlations rather than regression analysis here, given the risk of drawing misleading conclusions. I find the scatter plots valuable, as they provide a clear visual representation of the weak (but statistically significant) correlation between spring temperature and excess mortality.

Regarding monthly mortality data, I acknowledge that this type of data is largely unavailable for the period in question. While monthly data would likely provide additional insights, your approach with annual data remains meaningful, and clarifying this as a limitation while mentioning the future potential of monthly data is a good addition.

One area that might benefit from further detail is the interpretation of temporal patterns in Figure 6. While improving living conditions may account for some of the observed changes in correlations over time, a more explicit discussion of how broader socioeconomic factors – such as advancements in medical care, shifts in population density, nutritional improvements, and changes in public health infrastructure – could influence these mortality trends might enrich the analysis. However, I do not insist on this, as the revisions already made add further insight.

Additionally, I appreciate the expanded discussion on regional vulnerability patterns, specifically addressing why areas with milder winters (paradoxically) might be more vulnerable to cold-related mortality.

Overall, I am pleased with the planned revisions and look forward to seeing the published version of the article.